# Introducing principles of synaptic integration in the optimization of deep neural networks

Giorgia Dellaferrera [1,2✉], Stanisław Woźniak [1], Giacomo Indiveri [2], Angeliki Pantazi[1] & Evangelos Eleftheriou[1,3]

Plasticity circuits in the brain are known to be influenced by the distribution of the synaptic weights through the mechanisms of synaptic integration and local regulation of synaptic strength. However, the complex interplay of stimulation-dependent plasticity with local learning signals is disregarded by most of the artificial neural network training algorithms devised so far. Here, we propose a novel biologically inspired optimizer for artificial and spiking neural networks that incorporates key principles of synaptic plasticity observed in cortical dendrites: GRAPES (Group Responsibility for Adjusting the Propagation of Error Signals). GRAPES implements a weight-distribution-dependent modulation of the error signal at each node of the network. We show that this biologically inspired mechanism leads to a substantial improvement of the performance of artificial and spiking networks with feedforward, convolutional, and recurrent architectures, it mitigates catastrophic forgetting, and it is optimally suited for dedicated hardware implementations. Overall, our work indicates that reconciling neurophysiology insights with machine intelligence is key to boosting the performance of neural networks.

[1] IBM Research - Zurich, Rüschlikon, Switzerland. [2] Institute of Neuroinformatics, University of Zurich and ETH Zurich, Zurich, Switzerland. [3]Present address: Axelera AI, High Tech Campus 5, 5656 AE Eindhoven, Netherlands. ✉email: gde@zurich.ibm.com

Artificial neural networks (ANNs) were first proposed in the 1940s as simplified computational models of the neural circuits of the mammalian brain[1]. With the advances in computing power[2], ANNs drifted away from the neurobiological systems they were initially inspired from and reoriented towards the development of computational techniques currently employed in a wide spectrum of applications. Among the variety of techniques proposed to train multi-layer neural networks, the backpropagation (BP) algorithm[3,4] has proven to lead to an effective training scheme. Despite the impressive progress of machine intelligence, the gap between the potential of ANNs and the computational power of the brain remains to be narrowed. Fundamental issues of ANNs, such as long training time, catastrophic forgetting[5], and inability to exploit increasing network complexity[6], need to be dealt with not only to approach the human brain capabilities, but also to improve the performance of daily used devices. For instance, reducing the training time of online learning in robotic applications is crucial to ensure a fast adaptation of the robotic agent to new contexts[7] and to reduce the energy costs associated with training. Several techniques, such as *batch normalization*[8], *layer normalization*[9], and *weight normalization*[10], have been proposed to accelerate the training of ANNs. Although successful in improving the convergence rate, such methods are still far behind from the learning capabilities of the biological brain.

The limitations of ANNs with respect to the brain can be largely ascribed to the substantial simplification of their structure and dynamics compared to the mammals' neural circuits. Several mechanisms of paramount importance for brain functioning, including synaptic integration and local regulation of weight strength, are typically not modeled in BP-based training of ANNs. Overcoming this limitation could be key in bringing artificial networks' performance closer to animal intelligence[11]. Synaptic integration is the process by which neurons combine the spike trains received by thousands of presynaptic neurons prior to the generation of action potentials (APs)[12]. The axonal APs are elicited in the axon of the cell as a response to the input received from the cell's dendrites, and act as overall output signal of the neuron. Experimental evidence has shown that, at least in CA1 cells, input signals reaching the same postsynaptic cell from different presynaptic neurons may interact with non-linear dynamics, due to the active properties of the dendrites[13,14]. Specifically, when strong depolarization occurs in a dendritic branch, a dendritic AP is elicited in the region. The dendritic AP boosts the amplitude of the sum of the excitatory postsynaptic potentials (EPSPs) that generated it, thereby amplifying the dendritic input before it reaches the soma to elicit an axonal AP. The generation of a dendritic spike requires that enough presynaptic cells spatially connected to the same branch are active close in time with sufficient synaptic strength. As a consequence, the ability of synaptic inputs to influence the output of the postsynaptic neuron depends on their location within the dendritic tree. The differences between axonal and dendritic spikes are discussed in ref. [13]. The powerful computational abilities of neurons are suggested to stem from the complex nonlinear dynamics derived from dendritic spikes[15]. Figure 1a illustrates such a mechanism and shows how the impact of each presynaptic neuron depends also on the signals delivered to the same postsynaptic neuron through other presynaptic connections. Thus, the local weight distribution can be responsible of boosting the input signal at specific nodes. Similarly to neurons in the brain, nodes in ANNs receive inputs from many cells and produce a single output. We can relate the activation of artificial nodes to axonic APs, but there is no rigorous translation of the mechanism of dendritic APs into the dynamics of point neurons. However, dendritic spikes are strongly affected by the distribution of synaptic strengths within dendritic branches. Similarly, the non-linear dynamics of artificial nodes are affected by the weight distribution of synapses incoming to a layer of nodes. Surprisingly, in common training approaches for ANNs, a mechanism taking into account the weight distribution for each node is lacking.

Furthermore, synaptic plasticity in the brain is driven mainly by local signals, such as the activity of neighboring neurons[16]. The local interaction between synapses plays a crucial role in regulating weight changes during learning. In this context, the mechanism of heterosynaptic competition allows regulating synapse growth by limiting the total strength of synapses connected to the same pre- or postsynaptic neuron[17]. This phenomenon occurs as a nonlinear competition across synapses at each neuron. Specifically, as the summed weight of synapses into (or out of) a neuron hits a threshold, all the incoming (or outgoing) synapses to that neuron undergo a slight heterosynaptic long-term depression ("summed-weight limit rule")[18]. Additionally, in the cortex, each neuron tends to target a specific firing rate, and synaptic strengths are regulated to keep such rates constant, despite input perturbation[19]. Synaptic scaling acts as a global negative feedback control of synaptic strength, regulating the weight changes based on the neural local activities[20–22]. These homeostatic mechanisms are typically not modeled in the training of standard ANNs, which rely on global signals instead of local information[23,24]. Indeed, the BP algorithm relies on the simplified training strategy of assigning the error on a weight-by-weight fashion. Each synaptic weight is updated based on its *individual* contribution to the *global* output error of the network as a response to a specific input sample. We refer to this input-specific contribution as *input-driven responsibility*. Although earlier works have attempted to encode metaplasticity (i.e., the alteration of synaptic plasticity[25]) in the training of networks via weight-dependent learning rules[26–29], they invariably depend on a modulation of the Hebbian learning rule rather than ANNs training and do not account for the local weight distribution. Some training strategies more biologically plausible than BP[16], such as the feedback alignment (FA) algorithm[30] and its direct and indirect feedback alignment (DFA, IFA) variants[31], have been proposed, yet they do not explicitly model the neural mechanisms mentioned above.

Here, we make progress in reconciling neurophysiological insights with machine intelligence by proposing a biologically inspired optimizer that incorporates principles from biology, including synaptic integration, heterosynaptic competition[32], and synaptic scaling[19]. Our approach achieves substantial benefits in the training of fully connected neural networks (FCNNs), leading to a systematically faster training convergence, higher inference accuracy, and mitigation of catastrophic forgetting. Our novel approach effectively boosts also the performance of convolutional neural networks (CNNs) and spiking neural networks (SNNs)[33] on temporal data. These results validate the hypothesis that biologically inspired ANN and SNN models feature superior performance in software simulations[34], and provide guidelines for designing a new generation of neuromorphic computing technologies[35].

## Results

**The GRAPES algorithm.** The synaptic integration and the local synaptic strength regulation mechanisms are complex processes that depend on various factors, such as the large variability in size, structure, excitability, intercellular distance, and temporal dynamics of synapses and dendritic spines[36]. The simple point-like structure of a synchronously operating ANN node does not allow one to reproduce the rich dynamics enabled by the

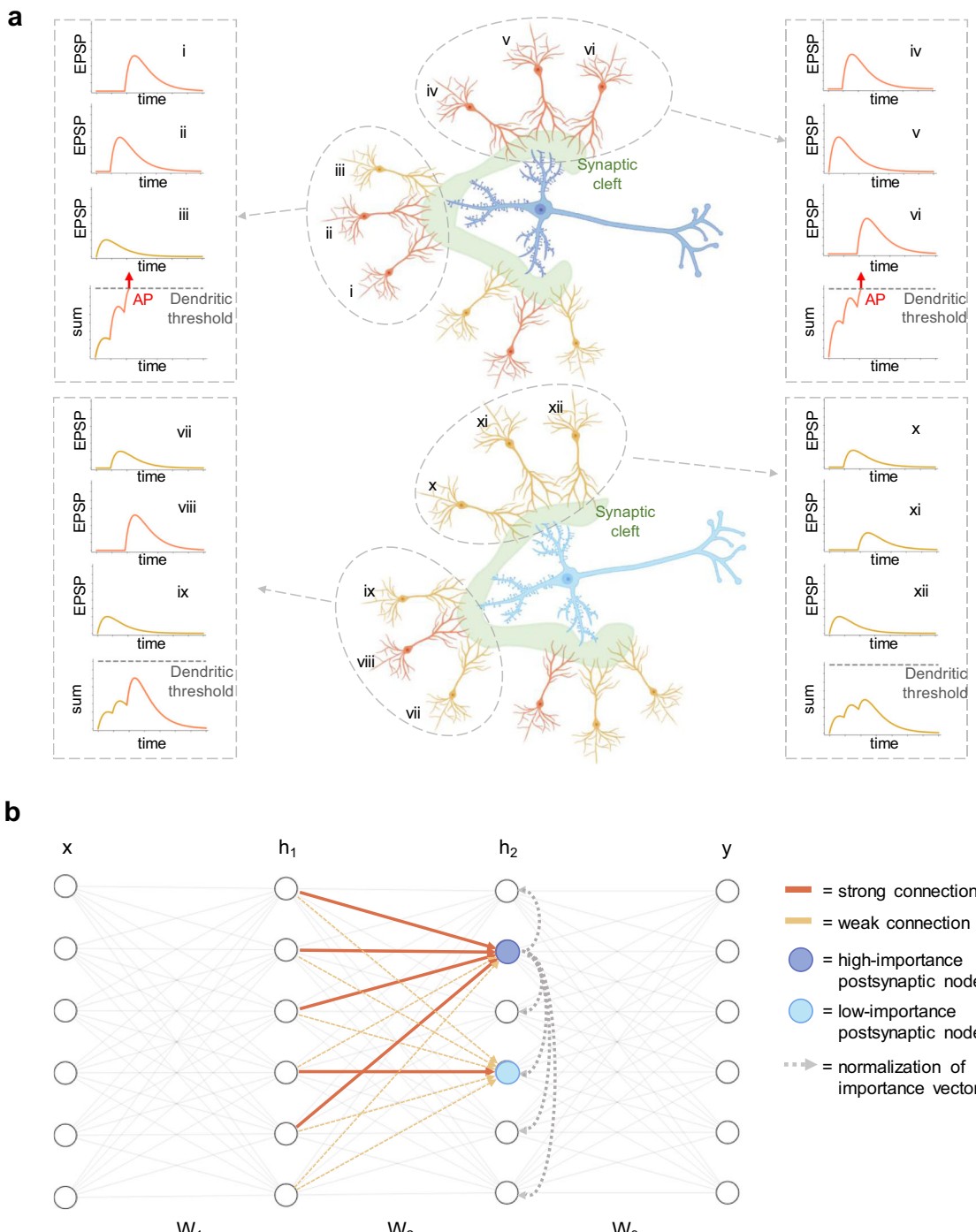

**Fig. 1 Synaptic strength distribution in biological and artificial networks. a** In biological synapses, during the process of synaptic integration, dendritic spikes can enhance the impact of synchronous inputs from dendrites belonging to the same tree. Excitatory postsynaptic potentials (EPSPs) with the same amplitude but different locations in a dendritic tree may lead to different responses. For example, dendrites *i, iv,* and *viii* send similar signals, but only the *i* and *iv* contribute in driving an AP, since their respective branches receive sufficient further excitation from other connected dendrites. In the top image, the postsynaptic neuron (dark blue) receives inputs mostly from dendrites generating strong EPSPs (orange) and only few generating weak EPSPs (yellow). The bottom postsynaptic neuron (light blue) receives most inputs from weak-EPSPs dendrites. Because of such a dendritic distribution, the dark blue neuron exhibits higher firing probability and thus its *importance* is higher with respect to the light blue neuron. **b** The structure of an FCNN is much simpler than that of biological neurons with presynaptic connections arranged in dendritic trees. However, analogously to panel (**a**), the *node importance* of each node arises from the distribution of the weight strength within each layer. The blue node has a high *node importance* since most of its incoming synapses are strong. Conversely, the light blue node's *importance* is lower, since the presynaptic population exhibits a weaker mean strength. The gray dotted lines emanating from the neuron with the highest importance and arriving at the other nodes in the same layer represent a normalization effect, resembling the winner-take-all competition based on the highest importance value.

neuronal complex morphology. Hence, a direct translation of the mechanism of dendritic integration for ANNs is not straightforward. Here, we take inspiration from the nonlinear synaptic dynamics and introduce a deep learning optimizer to boost the training of FCNNs. Our goal is to present an effective algorithm, inspired by biological mechanisms, and elucidate its potential impact on the properties of ANNs. This novel approach can also be easily applied to more biologically plausible neuronal models such as SNNs. Our algorithm builds on three observations:

(i) In the brain, due to the spiking nature of the information, a signal is propagated only if a postsynaptic neuron receives enough input current from the presynaptic population to elicit APs. In rate-based models of neural activity[37], a neuron with high firing rate is more likely to elicit high activity in the downstream neurons than neurons with low firing rate.

(ii) A single presynaptic neuron is responsible only for a fraction of the driving force that leads the postsynaptic neuron to fire. Hence, the impact of a presynaptic neuron on the downstream layers depends also on all the other presynaptic neurons connected to the same postsynaptic cell.

(iii) If we neglect specific distributions of the inputs, the firing probability of a postsynaptic neuron depends on the average strength of the presynaptic connections. If the average strength is high, the postsynaptic neuron is more likely to reach the spiking threshold and thus to propagate further the information encoded in the presynaptic population. Therefore, the postsynaptic neuron and the related presynaptic population have a high *responsibility* on the network's output and its potential error.

We refer to the intrinsic responsibility of the network as *network-driven responsibility*, as opposed to the *input-driven responsibility* mentioned above. Analogously, we introduce for ANNs the notion of *node importance* stemming from the node responsibility in propagating the information received from its presynaptic population to the output layer. The concept of *node importance* builds on the mechanism of dendritic integration in brain. In biological neurons, the dynamics of dendritic spikes originate from the spatial grouping of input cells based on the dendritic branch they send their signal to. In FCNNs the presynaptic nodes are connected to the postsynaptic nodes without being first grouped into dendritic branches. However, such a simpler non-dendritic morphology still offers the possibility to perform spatial grouping of input nodes based on the postsynaptic layer. In absence of dendritic branches, presynaptic cells can be grouped based on point-like postsynaptic nodes rather than on the dendritic branches of a single neuron. Consequently, while in biological circuits the dendritic integration is related to boosting input signals at the level of branches, the *node importance* in FCNNs is related to signal modulation at the level of point-like nodes. Therefore, the *node importance* is related to the average strength of the synapses connected to such node. Figure 1b illustrates the concept of *node importance* in an FCNN.

Based on these notions, we devised a novel algorithm, that we call GRAPES (Group Responsibility for Adjusting the Propagation of Error Signals). For simplicity, we begin by presenting the algorithm as a modulation of the error propagation in a network trained with BP and optimized with stochastic gradient descent (SGD). Next, we demonstrate that GRAPES can be conveniently applied also to other commonly used optimizers, such as Nesterov accelerated gradient (NAG)[38] and RMSprop[39], to other more biologically realistic training schemes, such as FA and DFA, and to networks with the biologically realistic dynamics of spiking neurons.

The GRAPES algorithm modulates the error signal at each synaptic weight based on two quantities: the *node importance* and the *local modulation factor*. Mathematically, we define the *node importance* of a given node $n$ belonging to layer $l$ as the sum of the absolute strength over all weights of layer $l$ whose postsynaptic neuron is $n$

$$i_n^l = \sum_{\text{pre}=1}^{N} \left| W_{n,\text{pre}}^l \right|, \tag{1}$$

where $N$ is the number of incoming connections to node $n$. The sum is performed over the absolute value of the parameters since we consider the connection strength (i.e., how much each weight amplifies or attenuates an incoming signal) rather than its excitatory or inhibitory nature. Alternatively, in specific cases discussed further in the paper, the *node importance* may also be obtained from the sum of the absolute strength of all weights of layer $l+1$ outgoing from the same presynaptic neuron. We remark that the importance defined in Eq. (1) depends only on the intrinsic state of the network and not on the input value. Therefore, in the initial phases of training, the importance is not related to the task. However, as the network undergoes training, the weight distribution becomes dependent on the task which the network is trained on, and, consequently, the importance vector evolves to be specific to the task.

The *importance vector* $i^l$ for layer $l$ contains the *node importance* values for each postsynaptic node $n$ in $l$. By normalizing the importance vector by its maximal value, multiplying it by 2, and lower-bounding by 1, we obtain the *local modulation vector*

$$m^l = \min\left(2\frac{i^l}{\max(i^l)}, 1\right), \tag{2}$$

whose elements are bounded in the range $1 \leq m^l \leq 2$. Such an interval has been defined on the basis of an empirical optimization (see Supplementary Note 1). The $n$th *local modulation factor* is the $n$th element of the resulting vector and indicates the responsibility of the postsynaptic node $n$ and its associated connected weights on the output of the network. In order to build the *local modulation matrix* $M^l$ for layer $l$, the *local modulation vector* is tiled as many times as the presynaptic population size. Each element of the matrix is associated with one synaptic weight of layer $l$. Therefore, by construction, the modulation has the same value for all weights $W_{n,\text{pre}}^l$ connected to the same postsynaptic neuron $n$.

With these quantities at hand, in the local version of GRAPES, we adjust the error signal in layer $l$ through a Hadamard multiplication of the weight-update matrix $\delta W_o^l$ computed with the standard optimizer (e.g., SGD) with the local modulation matrix $M^l$. The weight-change matrix, in which each row corresponds to a post-synaptic node and each column to a pre-synaptic node, is therefore modulated row-wise:

$$\delta W_M^l = \delta W_o^l \odot M^l. \tag{3}$$

The main steps of the computation of the importance vector, the local modulation vector, the matrix for a single hidden layer $l$, and the update step of the local version of GRAPES are summarized in Fig. 2a, b. The same concept of importance and modulation as described for the fully connected models can be applied to convolutional layers. In fully connected layers, the computation of the importance relies on grouping the weights (1D connections) based on the postsynaptic node. In convolutional layers, we compute the importance of each 2D filter by grouping the filters based on the postsynaptic maps. Supplementary Figure 1 shows a schematic of the main steps to compute the

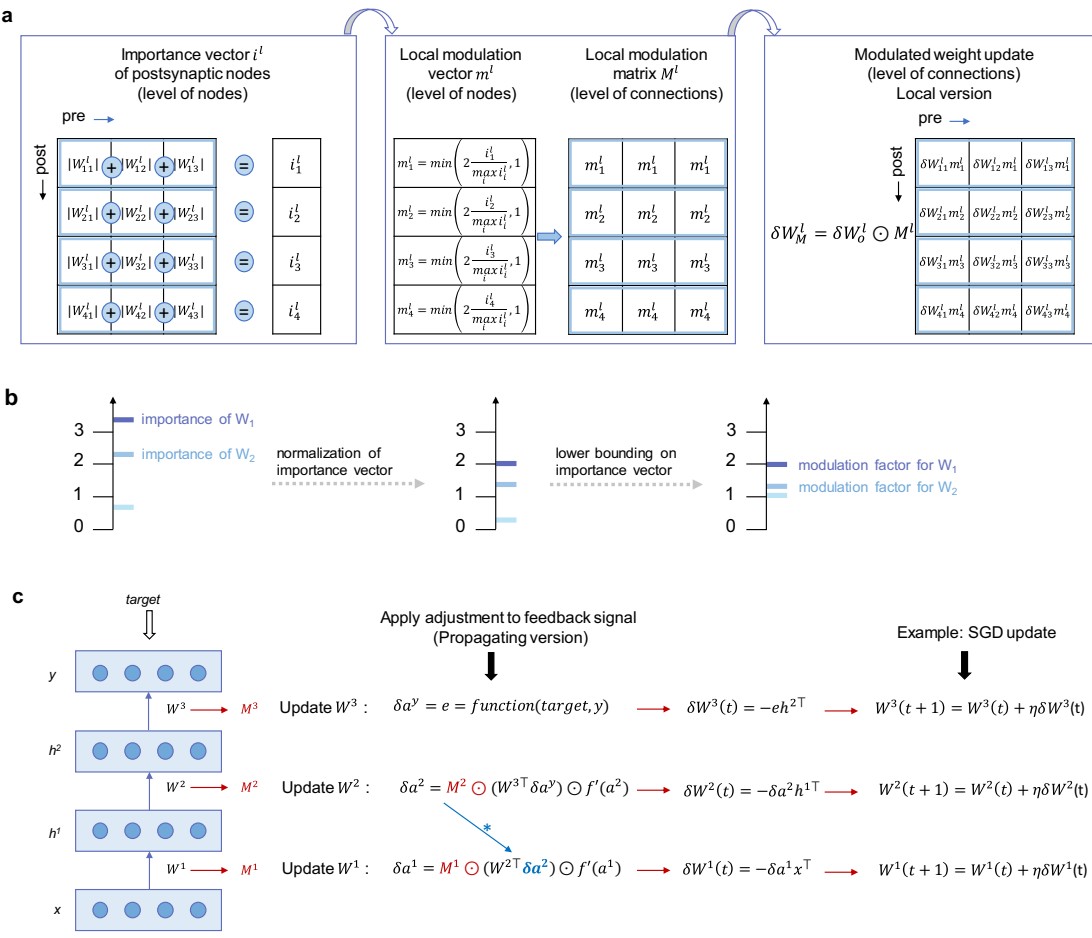

**Fig. 2 Computation of the modulation factor. a** The computation of the importance vector for one hidden layer, based on the associated weight matrix, is followed by the computation of the local modulation vector, based on the node importance. In the local version, GRAPES adjusts the weight update through the Hadamard multiplication of the initial weight-update matrix $\delta W_o^l$ with the local modulating vector $M^l$. **b** Sketch illustrating the steps in computing the modulation factor using the importance vector in the simple case of three postsynaptic nodes. **c** Algorithm for the propagation of the modulation factor to the upstream layers in a two-hidden layer network. The activations in the network are computed as $a^1 = W^1x$, $h^1 = f(a^1)$ and $a^2 = W^2h^1$, $h^2 = f(a^2)$, and the network output as $a^y = W^3h^2$, $y = f_y(a^y)$. Note that $\delta a^2$ is adjusted through the modulation matrix $M^2$ and such adjustment also affects the upstream layer since $\delta a^1$ contains $\delta a^2$.

filter importance. The filter update is modulated accordingly as described in Eq. (3).

In the propagating version of the algorithm, the modulation factor is incorporated in the error signal of each layer and propagated to the upstream layers, where it is incorporated in the respective weight updates. In Fig. 2c we outline the algorithm for the propagation of the modulation factor in a two-hidden layer network. The propagating version provides the greatest benefits in terms of classification accuracy and convergence speed compared to the local version, as shown in the Supplementary Table 2. Hence, the propagating version of the algorithm is the default method adopted in the simulations. Finally, the weight update obtained by applying the GRAPES modulation to SGD can be expressed as:

$$W^l(t+1) = W^l(t) + \eta \delta W_M^l(t). \tag{4}$$

GRAPES does not change the computational complexity of SGD and, since the modulation factor is computed only after the parameter update (e.g., at the end of each batch), the additional computations are negligible for large batch size.

By construction, the error signal modulation implemented in GRAPES presents some analogies with the biological mechanisms of heterosynaptic competition and synaptic scaling. Firstly, the *node importance* is defined as the sum of the synaptic weights connected to the same node. As in heterosynaptic competition, the information on the total synaptic strength is used to modulate the weight magnitude. However, while in heterosynaptic competition the total synaptic weight is used to solely determine depression by changing directly the weights[18], in GRAPES the total weight information is used to adjust the weight update, hence leading to both strengthening or weakening of the synapses. Secondly, by definition, the *local modulation factor* in GRAPES is equal for all synapses incoming to the same node. This leads back to synaptic scaling, in which a neuron-specific multiplicative scaling factor adjusts the weights of the synapses connected to the same neuron based on the local activity so that the neuron maintains a target firing rate.

Figure 3a displays the evolution of the mean of the modulation factor during training for each layer of a 10 hidden layer network. The dynamics of the modulation factor are different for each layer. The mean of the modulation factor exhibits the most pronounced decay in the first three hidden layers, whereas it either decreases very slowly or remains constant in the downstream hidden layers. In Fig. 3b, we show the distribution of the modulation factors for each layer after training. In each layer, a subset of the modulation factors is equal to 1, due to the lower bounding operation in Eq. (2). The remaining values are distributed with mean and variance specific to each layer.

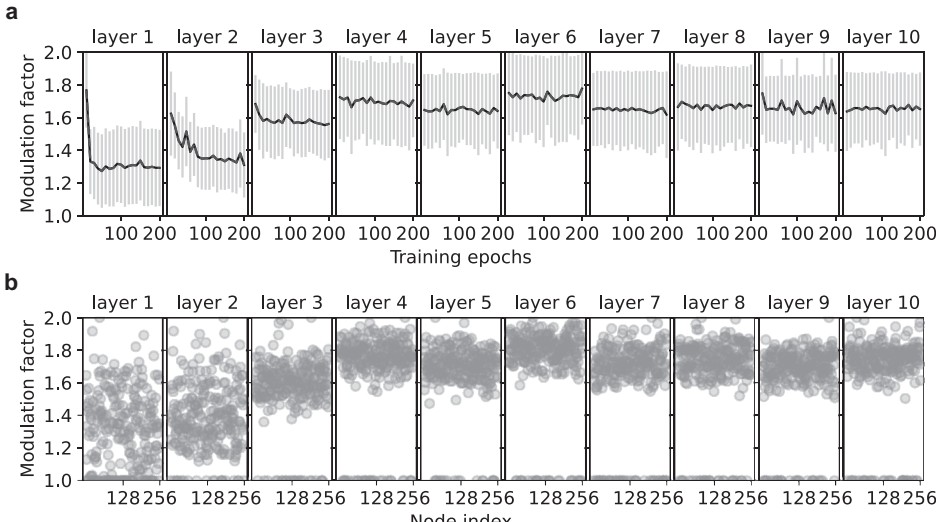

**Fig. 3 Dynamics of the modulation factor. a** Mean and standard deviation dynamics of the modulation factors for a 10 × 256 tanh network, with 10% dropout, trained with BP, SGD, and GRAPES modulation for 200 training epochs. The modulation factors are recorded for each layer every 10 epochs. **b** Distribution of the modulation factors at the end of training.

Therefore, based solely on the current state of the network weights, GRAPES offers a simple approach to modulate the error signal at each synapse using node-specific factors whose temporal evolution, mean and variance can be different for each hidden layer. Note that, for some layers, the decrease of the modulation factor with respect to its initial value (Fig. 3a) resembles a learning rate decay scheme[40]. Compared to the classical learning-rate-schedules, GRAPES provides two main advantages. First, to apply a learning rate decay, a time-consuming search of the best decay scheme and its hyperparameters is required for each network setting and task. Instead, the spontaneous decay provided by GRAPES (Fig. 3a) does not need to be optimized in advance, thus allowing the modulation factor to naturally adapt to different architectures and data sets. Furthermore, while with the conventional learning-rate-schedule approach the learning rate is equal for each parameter, GRAPES allows the update step to be adjusted differently for each weight (Fig. 3b). Specifically, we have shown in Fig. 3 that GRAPES implements a dynamic learning schedule for each weight. We demonstrate the stability of such a learning schedule by analytically proving the convergence properties of GRAPES in Supplementary Note 2.

**Simulation results on handwritten digit classification.** To illustrate the benefits of GRAPES on the training of ANNs, we have enhanced the standard minibatch SGD by incorporating the GRAPES modulation scheme, and are referring to it as "GRAPES". We initially compare the performance of GRAPES against standard minibatch SGD, which from now on we will simply call "SGD", on the MNIST data set[41].

To evaluate the convergence rate, we relied on a Michaelis Menten-like equation[42] and introduced the novel *plateau equation for learning curves*:

$$\text{accuracy} = \frac{\text{max\_accuracy} \cdot \text{epochs}}{\text{slowness} + \text{epochs}}. \qquad (5)$$

By fitting the test curve to this function, we can extract the *slowness* parameter, which quantifies how fast the network reduces the error during training. Mathematically, the *slowness* value corresponds to the number of epochs necessary to reach half of the maximum accuracy. Hence, the lower the *slowness*, the

faster the training. In our simulations, we perform the fit on the first 100 epochs. The graphical representation of the *plateau curve* is given in the Supplementary Fig. 2.

Figure 4a reports test curves and related *slowness* fits for 10 × 256 ReLU networks, trained on the MNIST data set. The red and blue curves refer to GRAPES and SGD-based training respectively, with the same learning rate $\eta = 0.001$. The testing curve for the GRAPES model saturates at a substantially higher accuracy plateau compared with those of the SGD models. Furthermore, the test curve for GRAPES rises much earlier and in a steeper manner—leading to a consistently smaller *slowness* parameter—compared with the test curves of the networks trained with SGD. This demonstrates that the key for improving the convergence lies in the non-uniform modulation of the error signal. Supplementary Table 3 shows that GRAPES exhibits the described improvements in accuracy and convergence rate under a wide range of network configurations.

Previous work in ref. [6] empirically showed that, as the number of trainable parameters in deep neural networks increases, the network performance initially improves and then saturates. We demonstrated that GRAPES is affected by performance saturation to a lesser extent than SGD. Figure 4b–d shows the accuracy results for models with increasing layer size, together with the corresponding slowness value. As the average of the modulation factor is larger than one (see Fig. 3), one needs to ensure that the training improvements are not solely due to a greater mean of the learning rate. Therefore, for each layer size, we perform a fine-grained learning rate search both for SGD and for GRAPES. We vary the learning rate from $\eta = 0.001$ to $\eta = 0.5$. Note that a further increase of the learning rate ($\eta > 0.5$) leads to instability and deteriorated accuracy. Figure 4c shows that for each learning rate and layer size pair GRAPES achieves better performance than SGD, with the improvements becoming increasingly more substantial as the learning rate is decreased. The optimal learning rate is $\eta = 0.1$ for SGD and $\eta = 0.05$ for GRAPES. With such values of $\eta$ we draw a cross section of the bar plots along the learning rate axis, as shown in Fig. 4b. We observe that for each layer size GRAPES outperforms SGD in terms of final accuracy. Moreover, the accuracy results indicate a rising trend for both SGD and GRAPES as the network layer size increases. Importantly, for GRAPES the rising trend saturates later than

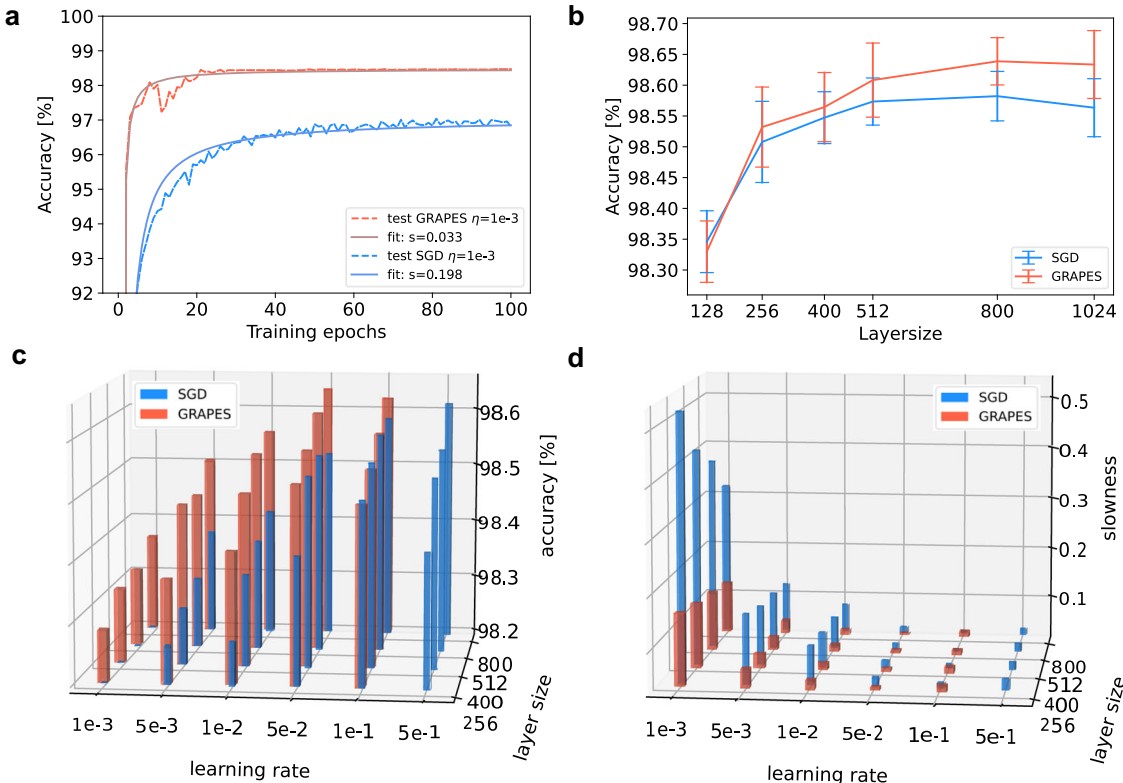

**Fig. 4 GRAPES applied to BP on MNIST.** Results of training fully connected models on MNIST with BP and SGD, with and without GRAPES. The red curves and bars correspond to the networks trained with GRAPES, whereas the blue curves and bars to the networks trained with SGD. **a** Testing curve (dotted line) and fit using the *plateau function* (solid line) for the 10 × 256 ReLU network, trained without dropout. The fit is performed on the initial 100 epochs of a single trial. **b** Test accuracy of networks with four hidden layers as a function of the layer size. The learning rate is optimized separately for SGD and for GRAPES. For different layer size, the optimal learning rate slightly varies. For most layer sizes the optimal learning rate is $\eta = 0.1$ for SGD and $\eta = 0.05$ for GRAPES. Each curve is a cross section of the bar plot in panel **c** along the learning rate axis. **c** Test accuracy of networks with four hidden layers as a function of the layer size and the learning rate, trained with 10% dropout. **d** Convergence rate of the models in (**c**). In panels **c**, **d** for visualization purposes the bases of the SGD and GRAPES bars are slightly shifted from each other. The actual learning rates and layer sizes are the same for both and are reported in the axes' labels. For panels **b**–**d** the accuracy for each run is computed as the mean of the test accuracy over the last 10 training epochs. The reported result is the mean and standard deviation (error bars in **b**) over the accuracy of 10 independent runs.

for SGD. Furthermore, as shown in the Supplementary Note 3, if a learning rate smaller than the optimized one is used, GRAPES shows a robust rising trend for different network depth and layer sizes, whereas with SGD the accuracy either saturates or deteriorates for increasing network complexity. In addition, as shown in Fig. 4d, GRAPES exhibits a much faster convergence compared with SGD. Unlike SGD, especially for small learning rate values, GRAPES benefits from a greater network complexity and converges even faster when deeper networks are used, indicating that GRAPES enhances the most relevant weight updates. Therefore, GRAPES not only improves the model performance and scalability, but it also provides a useful tool to mitigate issues such as lower accuracy and slower convergence rate that arise when the learning rate is not carefully optimized.

**Performance under various learning rules and data sets.** GRAPES can be combined with a wide range of momentum-based optimizers (e.g., NAG, rmsprop) and credit assignment strategies (e.g., FA, DFA, IFA). When combined with DFA, the computation of the modulation factor requires a modification. Since in DFA the propagation of the error occurs directly from the output layer to each hidden layer, the dimensionality of the error terms is different with respect to BP. Therefore, in order to incorporate the modulation factor in the error term, we compute

the *importance* based on the presynaptic grouping

$$i_n^l = \sum_{\text{post}=1}^{K} \left| W_{\text{post},n}^{l+1} \right|, \qquad (6)$$

where $K$ is the number of outgoing connections from node $n$ in layer $l$. We point out that, while DFA solves the weight transport problem[31], the propagating version of GRAPES requires to propagate the modulation factors from the output to the input layer, thereby still incurring in the weight transport requirement.

Figure 5 shows the improvements obtained by GRAPES in terms of accuracy and convergence rate when applied to FA and DFA over two data sets more challenging than MNIST: Fashion MNIST[43] and Extended MNIST[44]. We test networks with increasing layer size and with varying learning rate. We demonstrate that, when the optimal learning rate is used, for each model size GRAPES yields a better accuracy than SGD. Furthermore, similarly to BP, FA shows a rising trend of performance for increasing layer sizes. However, this trend was not observed with DFA. In terms of convergence rate, GRAPES combined with FA strongly mitigates the degradation for small learning rates. With DFA, GRAPES has a better slowness value than SGD for almost all models, both for large and small learning rates. Supplementary Table 4 shows that comparable improvements in accuracy and slowness are obtained under a wide range

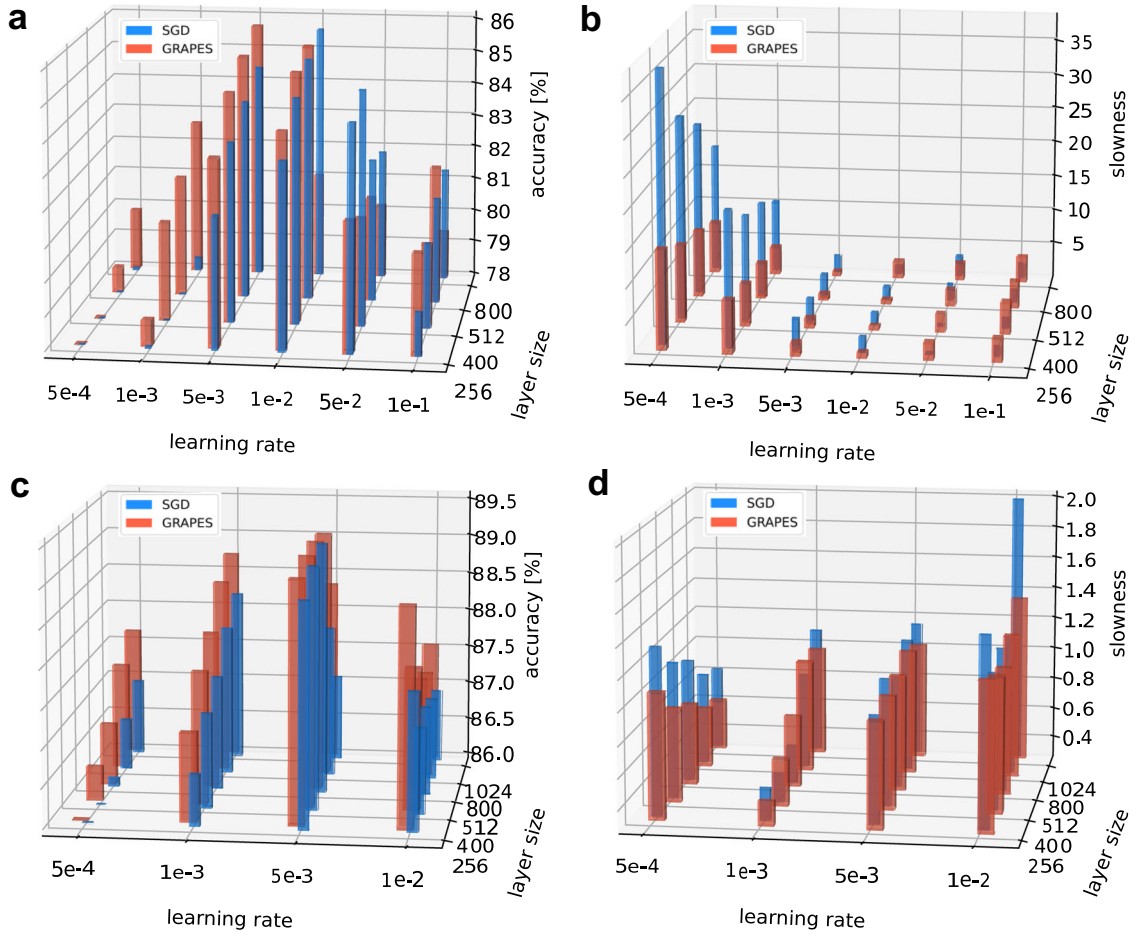

**Fig. 5 GRAPES applied to FA and DFA on Extended and Fashion MNIST.** Test accuracy and convergence rate in terms of slowness value for three-hidden layer ReLU networks, with 10% dropout, trained with FA on the Extended MNIST and DFA on the Fashion MNIST dataset, as a function of the layer size and the learning rate. The slowness parameter is computed by fitting the initial 100 epochs. The accuracy for each run is computed as the mean of the test accuracy over the last 10 training epochs. The reported result is the mean over the accuracy of ten independent runs. **a** Test accuracy and **b** convergence rate for FA on Extended MNIST. **c** Test accuracy and **d** convergence rate for DFA on Fashion MNIST. For visualization purposes, the bases of the SGD and GRAPES bars are slightly shifted from each other. The actual learning rates and layer sizes are the same for both and are reported in the axes' labels.

**Table 1 Test accuracy on the CIFAR-10 and CIFAR-100 datasets for CNNs trained with BP and Adam, with and without GRAPES modulation.**

| Optimizer | CIFAR-10 | CIFAR-100 |
|---|---|---|
| Adam | 84.78 ± 0.20 | 58.20 ± 0.39 |
| Adam+ GRAPES | **85.59 ± 0.17** | **58.85 ± 0.38** |

The network is a nine-layer residual architecture. The learning rate is decayed by 90% every 50 epochs and the initial learning rate is $\eta = 1e - 2$. The models are trained for 250 epochs. The accuracy for each run is computed as the mean of the test accuracy over the last 10 training epochs. The reported result is the mean and standard deviation over the accuracy of ten independent runs. The bold font indicates the best performance for each dataset.

of network settings. Furthermore, in Supplementary Tables 5–8 we report the performance of GRAPES applied on several feedforward models proposed in ref. [31] and ref. [45]. These results demonstrate that when GRAPES is applied on top of momentum-based optimizers, in most cases it leads to better accuracy than such optimizers in their original formulation.

Finally, we test the performance of the convolutional version of GRAPES on residual networks[46] trained on CIFAR-10[47] and CIFAR-100[48]. Specifically, we train a residual nine-layer architecture similar to[49] with the Adam optimizer, with and without GRAPES. Neither weight decay nor weight normalization is

applied. When GRAPES is used, the modulation is applied after the batch normalization and the nonlinearity. The learning rate is optimized separately for Adam and GRAPES (see Supplementary Table 9). The results are reported in Table 1. The best accuracy per dataset is reported in bold. Overall, GRAPES combined with Adam outperforms the standard Adam optimizer also on a residual architecture.

**Mitigation of catastrophic forgetting**. Catastrophic forgetting refers to the phenomenon affecting neural networks by which the process of learning a new task causes a sudden and dramatic degradation of the knowledge previously acquired by the system[5]. This represents a key limitation of ANNs, preventing the successful reproduction of continual learning occurring in the human brain[50,51]. Some of the most successful proposed approaches that enable lifelong learning rely on replay. This scheme involves fine-tuning models with old inputs[52] or their related compressed representations[53]. While replay is a biologically plausible mechanism, its application to the training of ANNs introduces additional computational steps and modifies the input sequence to include information about previous instances. Here, we show that the application of GRAPES mitigates, to a certain extent, the effects of catastrophic forgetting in ANNs without introducing additional steps or replaying data of previous instances.

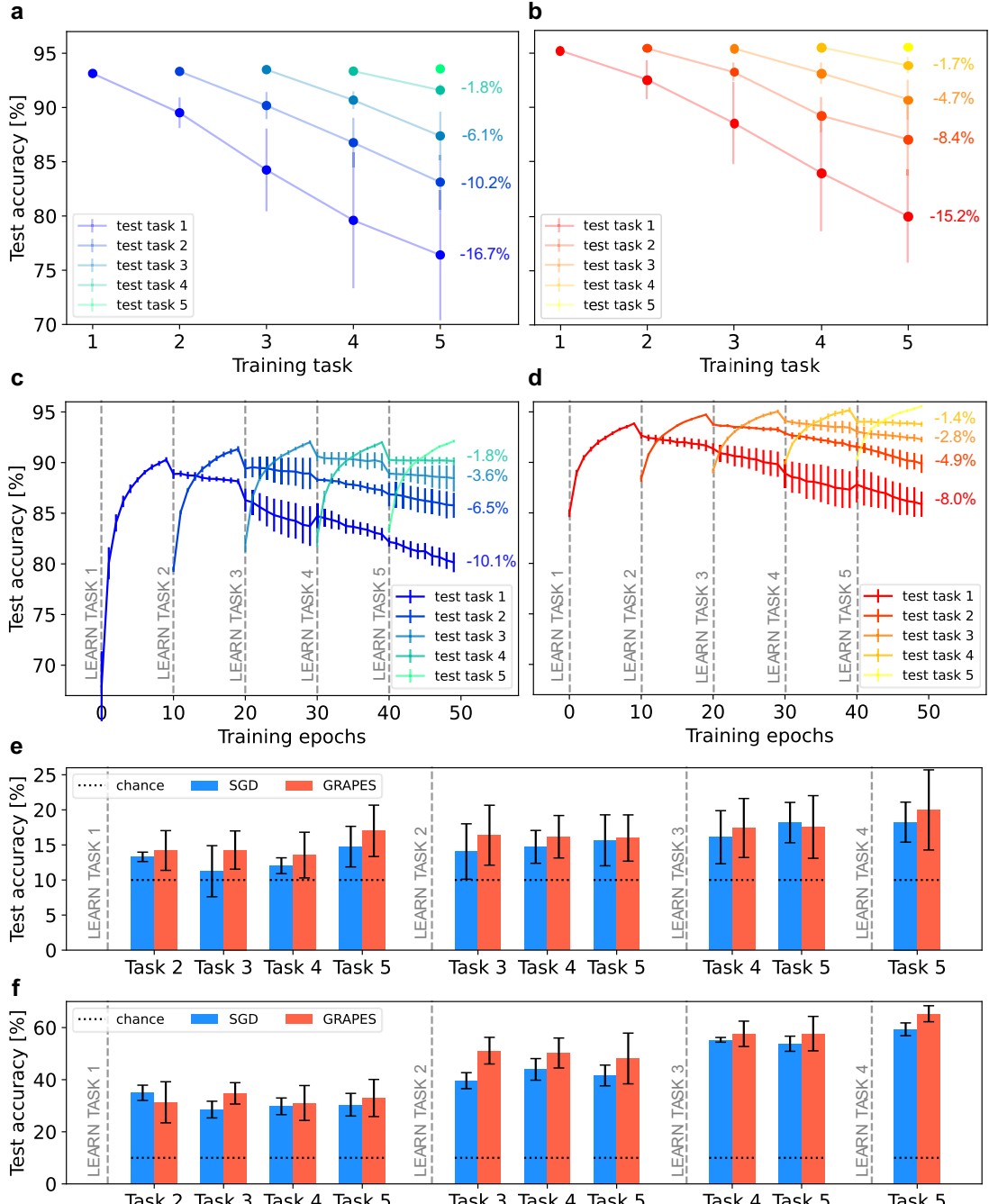

**Fig. 6 Mitigation of catastrophic forgetting.** Following the protocol in ref. [54] for catastrophic forgetting, we train ReLU networks with SGD with and without GRAPES on a sequence of tasks. Each task is defined by a random pattern of $N_p$ pixel permutations, which is applied to all MNIST train and test images. The networks are trained sequentially on each task for $N_e$ epochs. At the end of each training epoch, the networks are tested on both the task they are being trained on as well as the tasks they have already learnt (e.g., while learning task 1 they are tested only on task 1, and while learning task 2 they are tested on both task 2 and 1 to observe performance degradation on task 1). Panels **a** and **b** show the resulting test curves for $2 \times 256$ networks trained on the Avalanche benchmark with $N_p = 28 \times 28$ permutations (i.e., no overlap between tasks) and $N_e = 1$ epoch per task. For the optimization, we used SGD with momentum with default parameters. The results are mean and standard deviation over ten independent runs. Panels **c** and **d** show the resulting test curves for $3 \times 256$ networks trained with $N_p = 600$ permutations (i.e., small overlap between tasks) and $N_e = 10$ epochs per task. For the optimization, we used SGD without momentum. Panels **e** and **f** report the *per-task-future-accuracy* ([56]) on unseen tasks obtained with the same task setup as in (**c**) and (**d**). The tasks are defined by $N_p = 600$ (panel **e**) and $N_p = 300$ (panel **f**) pixel permutations, respectively. The networks are first trained on each task for $N_e = 10$ epochs and then tested on all the unseen tasks (*e.g.*, after learning task 1, the per-task-future-accuracy is reported for unseen tasks 2, 3, 4, 5).

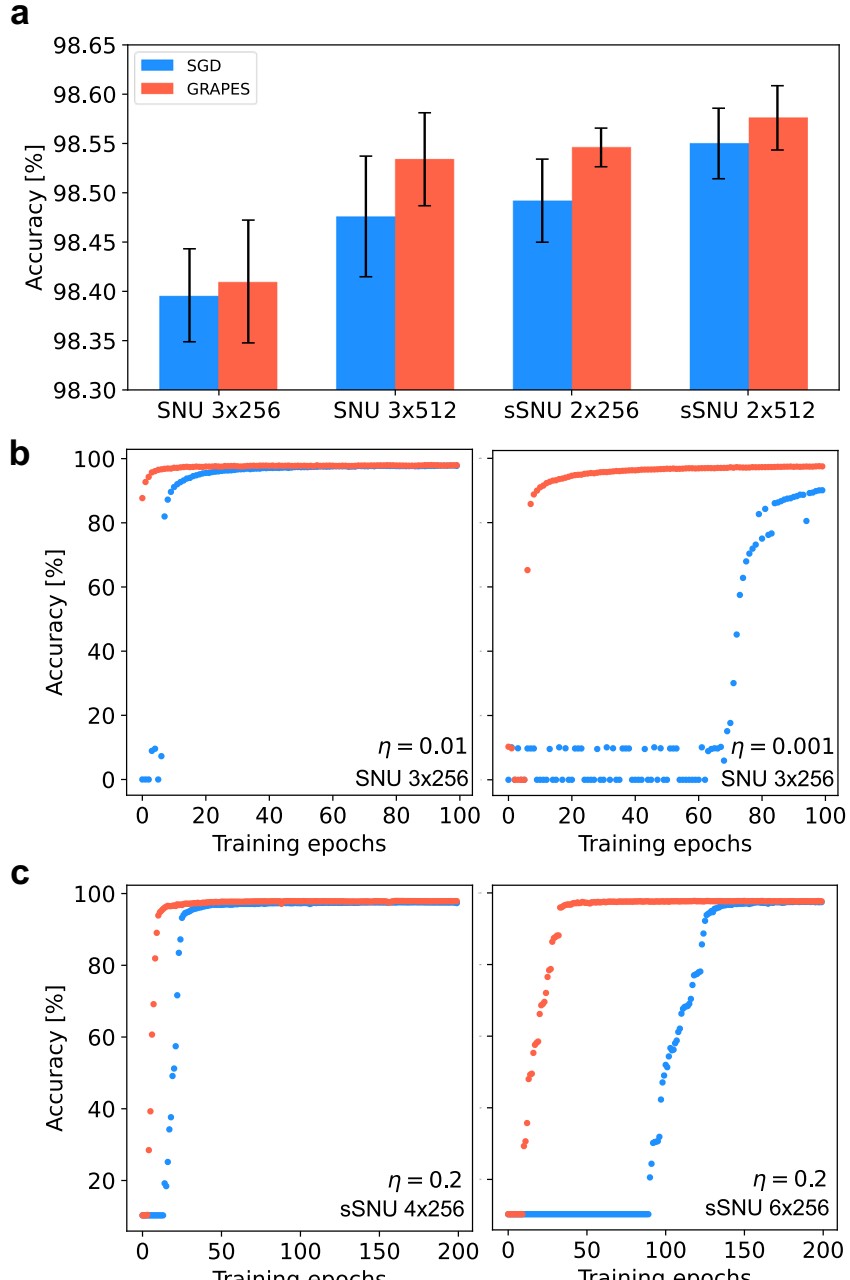

**Fig. 7 Experiments with spiking neural networks. a** Test accuracy for SNU and sSNU networks with three and two hidden layers, respectively. The results are mean and standard deviation of the test accuracy after training over five independent runs. **b** Test curves for SNU 3 × 256 networks with decreasing values for the learning rate $\eta$. Left hand-side: $\eta = 0.01$. Right hand-side: $\eta = 0.001$. The initial 100 training epochs are shown. **c** Test curves for sSNU networks with increasing number of hidden layers. Left hand-side: 4 × 256 networks. Right hand-side: 6 × 256 networks. In (**b**) and (**c**) the curves correspond to a single run.

To analyze catastrophic forgetting in a sequence of supervised learning tasks, we have adopted the protocol proposed in ref. [54]. For each task, we randomly generate a permutation pattern of a fraction of image pixels and we apply it to all the training and testing samples of the MNIST data set. We perform the training sequentially for all tasks for a fixed number of epochs, and, after each training epoch, we test the network performance on all the previously learnt tasks. First, we follow the protocol for the permuted MNIST proposed in the standardized benchmark Avalanche[55], which involves the permutation of all image pixels, therefore no overlap is present among tasks (Fig. 6a, b). Then, we use a custom task setup in which we regulate the fraction of permuted pixels per task, thereby exploring how GRAPES can exploit overlapping pixels among tasks. For each task, 600 random permutations are applied (see Fig. 6c, d). Both without and with overlapping pixels between the tasks, we observe that, compared with SGD, the drop in accuracy observed after learning each new task is considerably reduced when GRAPES is applied. In the latter case, the fraction of overlapping pixels among tasks reduces the accuracy degradation and its variability across different runs. Furthermore, the models, both with SGD and GRAPES, are able to exploit information from previous tasks, such that for each new task the test accuracy after one epoch is higher than for the previous tasks. Such accuracy is always better

for GRAPES than for SGD. We used the same learning rate for GRAPES and SGD ($\eta = 0.001$). If a larger learning rate is used, the performance degradation of SGD with respect to GRAPES further worsens (see Supplementary Fig. 3). We remark that the average learning rate for GRAPES is higher than that of SGD, so we could expect a higher performance degradation with respect to SGD. However, the importance-based modulation of the updates mitigates catastrophic forgetting more effectively than a uniform change of the learning rate.

Furthermore, we analyzed the effect of GRAPES under a second interesting aspect of incremental learning: the generalization to unseen tasks. Following the paradigm proposed in ref. [56], we compute the *per-task-future-accuracy*, by testing the model performance on tasks it has not been trained on yet. We initially used the same protocol with 600 permutations as in Fig. 6c, d, hence different tasks are only slightly correlated with each other (the total number of pixels is 784). Figure 6e shows that in most cases the networks trained with GRAPES show better generalization capability on unseen permutations. The absolute accuracy is, though, very low. Thus, we decreased the number of permutations to 300, leading the tasks to have a stronger correlation with each other. The results are reported in Fig. 6f. We observe that both SGD and GRAPES achieve well above-chance level accuracy. The generalization capability increases with the number of tasks learnt. Consistently with the results obtained with 600 permutations, GRAPES in most cases leads to higher accuracy than SGD. Therefore, both in the case of almost uncorrelated and partially correlated tasks, GRAPES proves to be more effective in achieving knowledge transfer to future tasks compared with SGD. We ascribe this remarkable result mainly to two properties of GRAPES. First, GRAPES enhances the updates related to a subset of parameters based on their importance. At each new task, such subset may vary, thus the learning focuses on different groups of synapses, thereby better-preserving knowledge on the old tasks. Secondly, as shown in Supplementary Note 4, the weights learnt with GRAPES are distributed with a larger variance in comparison to the weights learnt with SGD. We speculate that such a distribution might be more robust to performance degradation when the network is trained on a sequence of tasks.

**Application of GRAPES to biologically inspired neural networks.** SNNs are neural network models that attempt to mimic the complex neuronal dynamics of the mammalian brain[33]. Moreover, the development of SNNs is driven by the ultimate goal to implement embedded neuromorphic circuits, with high parallelism, low-power consumption, fast inference, event-driven processing, and online learning[57,58]. Given its biological inspiration, GRAPES holds great potential to boost the performance of SNNs. We apply GRAPES on SNN architectures implemented through the spiking neural unit (SNU) approach[59], which unifies SNNs with recurrent ANNs by abstracting the dynamics of a LIF spiking neuron[60] into a simple recurrent ANN unit. SNUs may operate as SNNs, with a step function activation, or as more conventional RNNs, with continuous activations. The non-spiking variant is called soft SNU (sSNU).

We train both SNU and sSNU models on temporal data derived from the MNIST data set. To that end, we encode the MNIST handwritten digit examples into spikes using the rate coding method as described in[59]. The depth of the network for optimal performance is found to be three hidden layers for SNU, and two hidden layers for sSNU. Figure 7a reports the accuracy results. For both models, GRAPES surpasses the classification accuracy obtained with SGD for different layer sizes. Furthermore, GRAPES renders the networks robust against

hyperparameter choice and model complexity. As can be seen in Fig. 7b for SNUs, the convergence of SGD-based training is heavily affected by changes in the magnitude of the learning rate $\eta$. As $\eta$ is decreased, the number of training epochs needed to trigger efficient learning dramatically rises. When GRAPES is introduced, the model reaches well above-chance performance in only a few epochs. Furthermore, as illustrated for sSNUs in Fig. 7c, SGD struggles in triggering learning of networks with increasing depth, requiring almost 100 epochs to start effective training of six hidden layer networks. GRAPES overcomes this issue, by enabling the deep models to converge with a lower number of epochs.

## Discussion

Inspired by the biological mechanism of non-linear synaptic integration and local synaptic strength regulation, we proposed GRAPES (Group Responsibility for Adjusting the Propagation of Error Signals), a novel optimizer for both ANN and SNN training. GRAPES relies on the novel concept of *node importance*, which quantifies the responsibility of each node in the network, as a function of the local weight distribution within a layer. Applied to gradient-based optimization algorithms, GRAPES provides a simple and efficient strategy to dynamically adjust the error signal at each node and to enhance the updates of the most relevant parameters. Compared with optimizers such as momentum, our approach does not need to store parameters from previous steps, avoiding additional memory penalty. This feature makes GRAPES more biologically plausible than momentum-based optimizers, as neural circuits cannot retain a substantial fraction of information from previous states[61].

We validated our approach with ANNs on five static data sets (MNIST, CIFAR-10, CIFAR-100, Fashion MNIST and Extended MNIST) and with SNNs on the temporal rate-coded MNIST. We successfully applied GRAPES to different training methods for supervised learning, namely BP, FA, and DFA, and to different optimizers, i.e., SGD, RMSprop, and NAG. We demonstrated that the proposed weight-based modulation leads to higher classification accuracy and faster convergence rate both in ANNs and SNNs. Next, we showed that GRAPES addresses major limitations of ANNs, including mitigation of performance saturation for increasing network complexity[6] and catastrophic forgetting[5].

We suggest that these properties stem from the fact that GRAPES effectively combines in the error signal information related to the response to the current input with information on the internal state of the network, independent of the data sample. Indeed, GRAPES enriches the synaptic updates based on the *input-driven responsibility* with a modulation relying on the *network-driven responsibility*, which indicates the potential impact that a node would have on the network's output, independently on the input. Such a training strategy endows networks trained with GRAPES with the ability to achieve convergence in a lower number of epochs, as the training is not constrained to information depending only on the presented training samples. For the same reason, such networks present better generalization capability than SGD both when tested on the learnt tasks and when presented with unseen tasks in continual learning scenarios. In this context, we identify parallelism with plasticity types in the brain. The change in synaptic strength in response to neuronal activity results from the interplay of two forms of plasticity: homosynaptic and heterosynaptic. Homosynaptic plasticity occurs at synapses active during the input induction, thus is input-specific and associative, as the *input-driven responsibility*. Instead, heterosynaptic plasticity concerns synapses that are not activated by presynaptic activity and acts as an additional mechanism to stabilize the networks after homosynaptic

changes[62,63]. Therefore, similarly to the *network-driven responsibility*, heterosynaptic plasticity does not exhibit strict input specificity.

Our algorithm appears to have certain similarity with existing normalization schemes[64] and the Winner-take-all computational primitive[65]. However, as GRAPES relies on the concept of network-driven responsibility, its main computations are based on synaptic strength rather than on synaptic activity. In particular, the operations of summing the weight strength and the normalization in Eq. (2) require that neurons communicate their synaptic strength. We remark that the normalization operation is introduced to project the importance vector into a meaningful interval, in which values larger than 1 lead to enhancement of the weight updates, and smaller than 1 imply weakening of the updates. The same numerical results can be achieved by performing the vector normalization by dividing by the mean of the importance values (or the total sum of the importance values), and later multiplying by $2 \times \frac{mean(i)}{max(i)}$ (resp. by $2 \times \frac{sum(i)}{max(i)}$) rather than 2. Furthermore, previous work has contemplated the possibility that neurons communicate synaptic strengths. For instance, in[18], the authors propose the summed-weight limit rule for heterosynaptic long-term depression: when the summed weight of synapses into (or out of) a neuron exceeds a limit, all the incoming (or outgoing) synapses to that neuron are weakened. Such a mechanism implies that synapses communicate information about the values of the synaptic weights to the postsynaptic node, and such information is used to modulate the synaptic weights in a non-local fashion. A second example is the theory of retroaxonal signals and neural marketplace proposed in[66,67]. Experimental evidence suggests that neurons are capable of carrying retroaxonal signals through molecules known as neurotrophic factors, which can encode information both on synaptic strength and on its temporal derivative. Such information is used to promote or hinder the consolidation of synaptic weights' changes. The theory of the neural marketplace builds on the mechanism of retroaxonal signaling and proposes a model for how networks of neurons in the brain self-organize into functional networks. Both the neural marketplace theory and the GRAPES algorithm rely on the propagation of information about the weights and their changes, hence the two frameworks present several analogies. First, the retroaxonal signals control the plasticity of synapses by modulating the synaptic updates. Similarly, the importance vector is used in GRAPES to modulate the weight changes prescribed by BP. Secondly, the retroaxonal signals carrying information on weight strength and weight change travel slowly; similarly, the information in GRAPES is only applied after each batch. Third, both the information propagated through neurotrophin and the importance in GRAPES do not depend on gradients. Finally, the theory in[67] introduces the concept of *worth* of a cell, which measures the usefulness of the cell's output, and is defined as the worsening in network performance if the cell were to die. A cell is inactivated if all its incoming connections are zeroes, hence the worth of a cell is related to the strength of the incoming synapses to the cell. Therefore, the worth can be related to the concept of *node importance* in GRAPES. In conclusion, the underlying ideas of GRAPES are inspired by the concepts of node importance, error modulation, and communication of weight strength, which are supported by experiments investigating the role of dendritic integration, synaptic scaling and retroaxonal signaling. While the biological inspiration is grounded on these mechanisms, only the high-level concept of GRAPES-like plasticity modulation is compatible with plasticity modulation principles observed in neural circuits.

The benefits of GRAPES stem from the adjustment of the error signal. The nonuniform distribution of the modulation factor, combined with the propagation to upstream layers, allows GRAPES to greatly enhance a subset of synaptic updates during training. Hence, small groups of synapses are enabled to strengthen or weaken to a much larger extent than with SGD. From preliminary investigation, GRAPES appears to convey the network weights toward more biologically plausible distribution, specifically heavy-tailed distributions[68–71]. Additional details are provided in Supplementary Note 4. We suggest that the properties exhibited by GRAPES could stem from such a weight distribution. Ongoing work in our group is currently seeking a more comprehensive understanding of this phenomenon.

Remarkably, our results suggest that GRAPES offers a promising strategy for mitigating the performance degradation caused by hardware-related constraints, such as noise and reduced precision, as discussed in Supplementary Note 5. We highlight that these constraints reflect biological circuits in many aspects, as the synaptic transmission is affected by noise and the neural signal is quantized. Interestingly, GRAPES retains many similarities with biological processes. We, therefore, envision that the biological mechanisms underlying GRAPES may play a central role in overcoming the limitations associated with hardware-related constraints. Furthermore, we suggest that such brain-inspired features are at the origin of the benefits of GRAPES on biologically-inspired models. Indeed, we have demonstrated that GRAPES not only improves BP-based training of standard ANNs, but additionally boosts substantially the performance of networks trained with biologically plausible credit assignment strategies, such as FA and DFA, and networks relying on the dynamics of spiking neurons. Both the FA algorithms and the SNN models are crucial steps towards bridging biological plausibility and machine learning. However, at the present stage, they can only achieve a limited performance compared to ANNs trained with BP[58,72]. For instance, as shown in the Results section, both the FA and SNNs approaches suffer from lower accuracy and convergence rate compared with BP, and SNNs training is severely affected by changes in network complexity and hyperparameters. Thanks to an efficient modulation of the error signal which enhances the updates of the most important parameters, GRAPES reduces the impact of such limitations, thereby narrowing the gap between the performance of bio-inspired algorithms and standard ANNs.

To conclude, our findings indicate that incorporating GRAPES and, more generally, brain-inspired local factors in the optimization of neural networks paves the way for pivotal progress in the performance of biologically inspired learning algorithms and in the design of novel neuromorphic computing technologies.

## Methods

**MNIST data set**. We train FCNNs with three and ten hidden layers, each consisting of either 256 or 512 hidden nodes. The activation functions chosen for the hidden layers are rectified linear unit (ReLU) or hyperbolic tangent (tanh). The output activation is softmax with cross-entropy loss function. With ReLU hidden nodes the weights are initialized according to[73], with tanh units according to[74]. The batch size is fixed to 64. The learning rate $\eta$ is optimized for the different models, separately for SGD and GRAPES, and is kept fixed during training. Supplementary Table 10 reports the optimized learning rate as well as detailed simulation settings for all simulations. We investigate the performance both without dropout[75] and with moderate dropout rates of 10 and 25%. We also show the accuracy improvement of the models that are trained with an augmented version of the data set, built by applying both affine and elastic deformation on the training set, similarly as proposed in ref. [76]. To compute the accuracy, we train the models on the training set and after each epoch we test the performance on the test set. Following the strategy in ref. [76], we report the best test accuracy throughout the entire simulation. For all settings, we average the result over five independent runs.

**Scalability to complex networks**. We train networks with layer sizes ranging from 128 to 1024 and with depth from 2 to 12 hidden layers. Each network is trained with ReLU, 10% dropout rate, and for 200 epochs. The learning rate is kept

constant to $\eta = 0.001$. As in the previous section, we report the best testing accuracy obtained throughout the entire training, averaged over five runs.

**Catastrophic forgetting**. In the simulations based on the Avalanche library, the networks, containing $2 \times 256$ ReLU hidden layers, are trained with momentum for one epoch on each task. In our task setting, for each task, 600 random permutations are applied. We train $3 \times 256$ ReLU FCNN networks on the training samples using shuffling and minibatch processing for a fixed number of epochs. We set the number of training epochs per task to 10. In both settings, we use a constant learning rate $\eta = 0.001$. We introduce a dropout rate of 10%. For all settings, we average the result over five independent runs.

**Spiking neural networks**. We train SNU and sSNU networks on the rate-coded MNIST data set. The dynamics of the units and the training protocol are the same as described in ref. [59]. The only difference with respect to the original SNU network in ref. [59] is the introduction of a soft-reset to smoothen the training process of the spiking units. We performed a grid-search for the hyperparameters. For the SNU networks the optimal configuration is three-hidden layer and constant learning rate $\eta = 0.1$. For sSNU models the optimal configuration is two-hidden layers with constant $\eta = 0.2$. The networks are trained for 200 epochs. The number of steps of input presentation is set to $N_s = 20$ during train and $N_s = 300$ during test. The mean and standard deviation of the final accuracy are computed over 5 runs.

**Programming**. The learning experiments of the ANN simulations were run using custom-built code in Python3 with the Numpy library. The SNU and sSNU-based simulations were performed using the original TensorFlow code from the Supplementary Material of[59].

## Data availability
The data sets used for the simulations are publicly available. Furthermore, source data are provided with this paper.

## Code availability
The program codes used for the numerical simulations are available at the repository https://github.com/IBM/GRAPES[77].

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

## Acknowledgements

We thank the reviewers for their many insightful comments and suggestions. We thank W. Senn, T. Bohnstingl, M. Dazzi, S. Nandakumar, A. Stanojevic, M. Pizzochero, L. Petrini, S. Dellaferrera, and our colleagues at the IBM Neuromorphic Computing and IO Links team for fruitful discussions. Figure 1 has been created with BioRender.com.

## Author contributions

G.D. conceived the idea. G.D., S.W., G.I., A.P., and E.E. identified the properties of the proposed algorithm in terms of error modulation, scalability, catastrophic forgetting, and behavior under hardware constraints. G.D. designed and performed the simulations. G.D., S.W., G.I., A.P., and E.E. analyzed the results. G.D. wrote the manuscript with input from the other authors.

## Competing interests

The authors declare no competing interests.
