## [Peer Review File · Nature Communications]

REVIEWER COMMENTS

Reviewer #1 (Remarks to the Author):

Summary

The authors present a method to normalize the weight update in a deep neural network by the sum of the weights over all incoming connections, a concept that is also observed in biological neural networks. Compared to vanilla stochastic gradient descent, this scaling results in faster convergence and better scalability across different datasets and strategies for credit assignment. Also, this method is shown to reduce catastrophic forgetting and shows similar benefits for spiking neural networks.

Strengths and Weaknesses

Strengths:

- This topic is relevant for a broad audience, at least to the field of machine learning and computational neuroscience.
- The presented methods are original and the main statements are supported by adequate ablation studies.
- The rigor and statistical evaluation fulfills the standards in the mentioned research fields.
- The methods is presented as simple and effective which makes it a good candidate for applications and further research studies.
- The manuscript is well structured and well written. The selection and quality of figures adequately support the main text.
- Raw results over multiple dimensions of the hyperparameter space are provided in the supplementary material.

Weaknesses:

- The experiments are limited to networks with fully-connected layers.
- The scalability of the method is theoretically addressed, but not actually shown for big networks (e.g. ResNet50) on large-scale datasets (e.g. ImageNet).
- The release of the source code upon publication would further improve the reproducibility.
- The mathematical notation is partly ambiguous.
- Thorough ablation study of the dependency on the learning rate is missing.
- Quality of figures, e.g. resolution of Fig. 3 and S3, could be improved.
- Reproducibility not clear, e.g. which Figure uses which dataset, network configuration, regularization, hyperparameters and number of trials? A table could be helpful, here.
- Evaluating on the best test accuracy does not comply with the best practices in machine learning.
- Please discuss related deep learning literature, e.g. [1].

The strengths significantly outweigh the weaknesses, such that I recommend the acceptance of this manuscript with minor modifications.

Detailed comments

- Fig. 1 and related main text: Please clarify the difference and relationship between axonal and dendritic spikes. In this context, please also discuss the relationship of these types of spikes to the activation of artificial point neurons.
- L136ff: Please clarify why more spikes carry more information and have a higher responsibility on the network's output and error. References are appreciated.

- Fig. 2: Where does the minus sign in $dW^i(t)$ come from?
- Fig. 2 last line: Are da^2 and da^1 swapped?
- Equation (1): Consider removing "post=".
- L197ff: Please elaborate on this empirical optimization. What happens if m is chosen outside of this interval?
- L234ff: Why does the computation of M after the parameter update reduce the computational cost?
- Fig. 4: The distribution of the data points (shaded area) does not exceed the interval defined by the standard deviation, which can mathematically not happen. Did you truncate the distribution?
- L321: A fine-grained grid-search of the learning rate for SGD and GRAPES would be appreciated.
- L453ff: Please elaborate how the difference of weight distributions is related to the number of epochs. I would not agree that the difference is necessarily smaller for fewer epochs.
- Fig. 7 b) right hand side: $n=0.001$ instead of $n=0.01s$?
- L551: I do not get the meaning of "relative", here.
- Suppl. Note 3: Not only the noise on and range of synaptic weights of hardware synapses is limited, but often also their weights, e.g. see [2]
- Table S3: Two results for GRAPES are missing.
- Table S7: Not clear what \pm denotes, standard deviation over trial averages?

[1] Salimans, T., & Kingma, D. P. (2016). Weight normalization: A simple reparameterization to accelerate training of deep neural networks. NIPS 2016.

[2] Pfeil, T., Potjans, T. C., Schrader, S., Potjans, W., Schemmel, J., Diesmann, M., & Meier, K. (2012). Is a 4-bit synaptic weight resolution enough?—constraints on enabling spike-timing dependent plasticity in neuromorphic hardware. *Frontiers in Neuroscience*, 6, 90.

Kind regards,
Thomas Pfeil

Reviewer #2 (Remarks to the Author):

Overview of Learning in Deep Neural Networks using a Biologically Inspired Optimizer:

The authors present an artificial neural network (ANN) optimization strategy that is inspired by learning that occurs in mammalian brains. In a nutshell, the authors modulate the backpropagation signal back through the ANN (each layer individually) by computing the "importance" that a single feature/node has on the learned task. Finding better, brain-inspired optimizers for machine learning is critical for the community as we incrementally work toward developing artificial general intelligence; however, the authors have not compared against recent state-of-the-art nor have they shown that GRAPES scales to larger problems. This makes it difficult for reviewers to measure the impact of the proposed work; and since this Nature Communications is a high-impact publication, I recommend rejecting the paper in its current form. Specific comments are below.

1. The authors need to elaborate how Equation 1 (the sum of the absolute value of the weights in a given layer "l" and a node "n") corresponds to importance for learning a given task.

2. In Equation 2, why is the modulation magnitude bound between 1 and 2? The authors say that it is empirical, but the authors should elaborate on this design choice.
3. In Equation 3, what is W_o ? I may have missed it, but I only see it defined in Figure 2a.
4. Figure 3 is confusing. It is not clear to me that the graphic provides any meaningful tie back to the main points made in Lines 258-293 of the text.
5. The experiments are "toy" examples that do not demonstrate the scalability of GRAPES. For example, on Line 407, the Lifelong Machine Learning (LML) community has replaced the MNIST data permutation experiment with larger scale problems such as ImageNet and Core50. I recommend checking out the REMIND paper located at <https://arxiv.org/pdf/1910.02509.pdf> for a scalable example, but there have already been improvements on this model.
6. The authors must compare their results against the state-of-the-art algorithms, which I believe is necessary for high-impact publications such as Nature Communications. REMIND listed on the bullet above is an example of that.
7. The LML Community is working to adopt a standardized benchmark located here: <https://avalanche.continualai.org/>. Especially for the toy examples, I strongly recommend using this framework to help other researchers provide apples-for-apples comparisons.
8. The supplementary material provides mean + standard deviation over a set of five runs. The author needs to provide statistical significance testing for the different methods and ablation studies used in this paper. This will help reinforce the importance of the proposed work, push the author to increase the sample size, or make modifications to GRAPES until statistically significant improvements are demonstrated.

Reviewer #3 (Remarks to the Author):

"Learning in Deep Neural Networks Using a Biologically Inspired Optimizer" reports a modification to the training procedure of artificial neural networks that integrates elements of a form of homeostatic plasticity observed in brain tissue. The modification is shown to lead to increased accuracy, greater learning speed and decreased vulnerability to catastrophic forgetting. Although a convergence proof is given in supplementary material, the main claims rely on computational simulations performed on benchmark machine learning tasks.

There is currently quite a lot of interest in spotting subtle features of real neuronal network that would alleviate some of the outstanding issues with training deep artificial neural networks. Despite this strong interest, successes are rare. Potent and truly bio-inspired improvements have been difficult to show. The present article is squarely situated in the focus of this lens, but for the achievement to be valid we have to be careful that the improvements observed arise from a fair comparison and that the modifications are truly biologically-inspired. My concerns are that these two points are not addressed in a convincing fashion in the present form of the manuscript.

Main concerns:

Fair comparison with SGD.

The results shown in Fig. 4 are impressive as the SGD and GRAPES show opposite dependence on depth. But I find this figure is an unusual description of SGD. Sure SGD can

lead to the sort of biphasic depth dependence shown in Fig. 4, but a monotonically increasing dependence such as that shown for GRAPES is certainly possible. For instance, Li and Sompolinsky have derived some conditions for when a depth increases vs decreases performance. A change in hyperparameter can change whether depth increase or decreases performance. There are quite a few hyperparameters here. Of note, the ones that were chosen empirically in the choice of the min/max function for defining m_l . It is easy to imagine that in switching from SGD to GRAPES, a switch in task statistics resulted such that depth became entirely beneficial. Importantly, SGD with a more elaborate meta-parameter search could also have given this improvement. The improvement may thus have to do more on the relative choice of hyper parameters than the training protocol. The description in methods being very scant on the hyperparameter search (it appears only the learning rate was systematically varied), one would need a better description for the rationale of the hyperparameter search and/or a proof that SGD is being treated fairly in that respect. Particularly given that SGD-based curves at odds with those in Fig. 4 have been reported many times in the literature. The results pertaining to slowness may be more robust as I don't know of any reports at odds with what is shown here (though it is seldomly investigated).

Similarly, the improvements shown in Fig. 5 and 6 are small. It becomes a question whether this improvement is robust to different network setups (on that note, it would be useful to define the error bars, many changes don't even seem to be significant as per these error bars). The results in Fig 6 in particular appear to be right in the error bars of one another and do not seem to have been statistically compared.

Biological inspiration. The present algorithm is inspired by an observed property of real synapses that when potentiating one synapse, other synapses onto the same neuron are depressed. Although this property is debated (see Beique et al. PNAS 2011), this type of constraint has long been an integral part of learning in biological neural network (see Oja's learning rule and PCA starting in 1982, up to more recent studies like Bird and Cuntz BioRxiv 2020). The approach taken here is different, however, as it does not normalize weights by the total input weight (as in Oja's rule), but by the *maximum* total input weight among the different units in a layer. As such, this introduces a form of non-local influence of synaptic weights. As if the neuron receiving the most weight is acting on all other neurons by dividing its total weight out of all synapse onto other neurons in the layer. To be specific, I am referring to Eq 2., which does look like the softmax function commonly used in conjunction with a reference to the biological winner-take-all computation implemented by feedback inhibition. But Eq 2 is on weights and neurons do not communicate weights so the feedback inhibition cannot apply. The authors mention that the same principle of homeostatic plasticity apply by considering Eq. 6, which is utterly confusing to me as I don't see how this is equivalent to Eq 2, nor how biology can do Eq 6. Together, I do not see how the core computational element studied here can be implemented by neurons.

Other criticism to improve manuscript.

Otherwise, I thought the manuscript structure difficult to follow.

The interwoven ideas of improving biological realism of ANNs and shining insight as to how to make biological simulations closer to ANNs was difficult to follow.

I was also confused by the two introductions.

The title is not specific as to what element is the focus of biological inspiration.

There were a number of inaccurate statements ('the stronger the input, the higher the

firing rate and thus the amount of information propagated to the next layer', 'incorporates key principles of synaptic integration observed in dendrites' (should be plasticity not integration)), and as hinted at in the above points, the links with existing literature both biological and ML literature could be incorporated better.

I did not feel that Fig 1a was communicating the main point of the training protocol. Fig 1b was missing the effect of Eq 2.

Reply to Reviewer #1's Comments:

1, Referee: *The authors present a method to normalize the weight update in a deep neural network by the sum of the weights over all incoming connections, a concept that is also observed in biological neural networks. Compared to vanilla stochastic gradient descent, this scaling results in faster convergence and better scalability across different datasets and strategies for credit assignment. Also, this method is shown to reduce catastrophic forgetting and shows similar benefits for spiking neural networks.*

Strengths:

- *This topic is relevant for a broad audience, at least to the field of machine learning and computational neuroscience.*
- *The presented methods are original and the main statements are supported by adequate ablation studies.*
- *The rigor and statistical evaluation fulfills the standards in the mentioned research fields.*
- *The methods is presented as simple and effective which makes it a good candidate for applications and further research studies.*
- *The manuscript is well structured and well written. The selection and quality of figures adequately support the main text.*
- *Raw results over multiple dimensions of the hyperparameter space are provided in the supplementary material.*

Response: We thank Reviewer #1 for the positive feedback and for highlighting the strengths of our work. Below is a point-by-point reply to the Reviewer's comments and suggestions. We have revised the manuscript accordingly.

2, Referee: *Weaknesses: - The experiments are limited to networks with fully-connected layers.*

Response: We have extended our work to convolutional networks by designing a version of the GRAPES optimizer for convolutional layers. We tested our algorithm on a residual convolutional network combined with the Adam optimizer and demonstrated that it improves the performance compared to the standard Adam scheme. We've described the slightly modified GRAPES algorithm for convolutional layers in the main text, *i.e.*, lines 266ff and provided a graphical representation of its main steps in Supplementary Figure 1. Finally, we've reported the simulation results in Table 1 of the manuscript.

3, Referee: *- The scalability of the method is theoretically addressed, but not actually shown for big networks (e.g. ResNet50) on large-scale datasets (e.g. ImageNet).*

Response: We thank Reviewer #1 for the suggestion to apply GRAPES on more complex scenarios than in the original manuscript. We have extended our work to a more complex model than the fully connected networks, namely a nine-layer residual model, and to a larger-scale dataset, specifically CIFAR-100. Table 1 described in the previous bullet point shows the new results.

4, Referee: *- The release of the source code upon publication would further improve the reproducibility.*

Response: We have included a "Code Availability" section in the manuscript, in which we have shared the link to the Github repository containing the code to reproduce the experiments (<https://github.com/IBM/GRAPES>).

5, Referee: *- The mathematical notation is partly ambiguous.*

Response: We thank Reviewer #1 for the comment. We have updated specific equations following the reviewer's detailed suggestions. We'd be happy to modify the mathematical notation and include any further specific suggestions.

6, Referee: - *Thorough ablation study of the dependency on the learning rate is missing.*

Response: We thank Reviewer #1 for the constructive criticism. We have expanded our study to include a fine-grained grid-search of the learning rate. To this end, we have systematically varied the learning rate in a range leading to successful training. In the revised Figure 4c, we show a 3d plot containing the performance of a fully connected network trained with backpropagation and SGD on the MNIST dataset, with and without GRAPES, for increasing layer size. For each layer size, we've trained the model with six learning rates, and showed that GRAPES outperforms the standard SGD for all the tested learning rate values. Such improvements are substantial for small learning rates, and they are also noticeable for the larger ones. Note that the best overall performance is achieved with the larger values of learning rates, in conjunction with the GRAPES method. We would like to point out that a further increase of the learning rate leads to instability and deteriorated accuracy. Similarly, we carried out an ablation study for the Feedback Alignment and Direct Feedback Alignment training schemes. In particular, we've focused the ablation studies on the Extended MNIST dataset for Feedback Alignment and on the Fashion MNIST dataset for Direct Feedback Alignment. The results are shown in the revised Figure 5a and 5c. For all the described scenarios, we've also analyzed the dependency of the slowness on the learning rate variation, and reported the results in the revised Figures 4d, 5b and 5d.

Furthermore, we have carried out an ablation study on the learning rate for the catastrophic forgetting experiments and for the SNU simulations. Regarding the catastrophic forgetting, Supplementary Figure 3 shows the results of the permutation experiment with a larger learning rate. Regarding the SNU, small changes in the learning rate did not affect the final performance of the models.

7, Referee: - *Quality of figures, e.g. resolution of Fig. 3 and S3, could be improved.*

Response: We thank Reviewer #1 for pointing out this limitation. We have improved the resolution of the figures in the revised manuscript.

8, Referee: - *Reproducibility not clear, e.g. which Figure uses which dataset, network configuration, regularization, hyperparameters and number of trials? A table could be helpful, here.*

Response: We thank Reviewer #1 for the suggestion. In the Supplementary Information we have included the Table 10 which reports the detailed settings for all the experiments.

9, Referee: - *Evaluating on the best test accuracy does not comply with the best practices in machine learning.*

Response: We thank Reviewer #1 for this remark. We have changed the evaluation metric accordingly. Specifically, in the revised Figures 4 and 5 the reported accuracy is the mean over the test accuracy computed after the last 10 training epochs, then averaged over 10 independent runs. In the experiments for catastrophic forgetting and robustness to hardware constraints, given the small number of training epochs, we've reported the test accuracy after the last training epoch. In the experiments with SNU the test accuracy is computed at the end of the training.

10, Referee: - *Please discuss related deep learning literature, e.g. [1].*

Response: We thank Reviewer #1 for the suggestion. We have included the reference in line 32 of the revised manuscript.

11, Referee: *The strengths significantly outweigh the weaknesses, such that I recommend the acceptance of this manuscript with minor modifications.*

Response: We thank Reviewer #1 for providing positive feedback and for recommending acceptance of the manuscript.

12, Referee: *Detailed comments - Fig. 1 and related main text: Please clarify the difference and relationship between axonal and dendritic spikes. In this context, please also discuss the relationship of these types of spikes to the activation of artificial point neurons.*

Response: We thank Reviewer #1 for the feedback. The diagram in Figure 7 in *Etherington et al. (2010)*, reproduced below, illustrates the differences between axonal and dendritic spikes. Dendritic spikes occur in proximity of the dendrites, as a response to the input signals received from presynaptic cells in different dendritic branches. Axonal spikes are generated close to the soma, as a response to the input signal that has already been integrated through different branches and has travelled along the axon, attenuating its amplitude. The dynamics of neurons with dendritic arborization features a greater complexity than the dynamics of point neurons, which do not model dendritic branches. While the activation of artificial nodes is related to axonic spikes, there is no rigorous translation of the mechanism of dendritic spikes into the dynamics of the node. However, dendritic spikes are strongly affected by the distribution of synaptic strengths within dendritic branches. Similarly, the nonlinear dynamics of artificial nodes are affected by the weight distribution of synapses incoming to a layer of nodes, leading to the concept of *importance* introduced in the manuscript. The novelty of our work lies in considering the distribution of the connection strength in the learning dynamics of artificial nodes.

In the revised manuscript we have clarified the relationship between axonal and dendritic spikes in lines 48ff, 55ff and their relationship with the activation of artificial nodes in lines 79ff.

Figure 7 Dendritic spikes, caused by regenerative opening of dendritic voltage-gated ion channels, can boost the somatic impact of synchronous synaptic inputs.

13, Referee: - L136ff: *Please clarify why more spikes carry more information and have a higher responsibility on the network's output and error. References are appreciated.*

Response: We thank Reviewer #1 for pointing out an unclear sentence. We've addressed the comment in lines 170ff of the revised manuscript. We would like to emphasize that our reasoning holds in the context of firing-rate based models, and thus we've modified the text to specify that neurons with higher firing rates are more likely to elicit higher activity in the downstream layers than neurons with low spiking rate. As reference, we have cited *Brette (2015)*, a review paper on rate-based and spike-time based theories.

14, Referee: - *Fig. 2: Where does the minus sign in $dW^i(t)$ come from?*

Response: We thank Reviewer #1 for pointing out a typo in Fig. 2. We've used the same notation as in *Nokland, NIPS 2016*, where the $dW^i(t)$ contains the minus sign. However, in the original Fig. 2 we had also a minus sign in the SGD update formula. We have now corrected this typo in the revised Fig. 2c and in Equation 4.

15, Referee: - *Fig. 2 last line: Are da^2 and da^1 swapped?*

Response: We thank Reviewer #1 for this comment. da^2 and da^1 are not swapped. We use the same notation as *Nokland, NIPS 2016*.

16, Referee: - *Equation (1): Consider removing "post=".*

Response: We have removed "post=" from Equation 1.

17, Referee: - L197ff: *Please elaborate on this empirical optimization. What happens if m is chosen outside of this interval?*

Response: We have extended our study to include different ranges for m . We've reported the new results in the Supplementary Note 1. As shown, the best empirical choice for the range of m is $[1,2]$, as in the original manuscript. However, note that the wider range $[1,3]$ leads to similar improvements with respect to the standard SGD.

18, Referee: - L234ff: *Why does the computation of M after the parameter update reduce the computational cost?*

Response: Since M is computed based on the weights of the models, it is sufficient to compute it only when the weights are updated. The computation of M represents an additional operation in the training. However, for a large batch size, the impact of the additional computational time on the overall training time is very small, as the computation of M is performed only once per batch.

19, Referee: - *Fig. 4: The distribution of the data points (shaded area) does not exceed the interval defined by the standard deviation, which can mathematically not happen. Did you truncate the distribution?*

Response: We thank Reviewer #1 for this comment. In the original Figure 4 we have used a violin plot to show the probability density of the points. In the revised Figure 4, we do not show the distribution of the data points as the revised figure is a surface plot.

20, Referee: - *L321: A fine-grained grid-search of the learning rate for SGD and GRAPES would be appreciated.*

Response: We thank Reviewer #1 for the useful suggestion. We've extended our study to include a fine-grained grid-search of the learning rate. Revised Figures 4 and 5 show the results for backpropagation on the MNIST dataset, and for feedback alignment and direct feedback alignment on the Extended and Fashion MNIST datasets, respectively. We've also performed a fine-grained grid-search for the experiments on catastrophic forgetting and we've found that, as expected, increasing the learning rate leads to a more severe performance degradation of both SGD and GRAPES. In this context, the improvements of GRAPES with respect to SGD are even more significant with larger learning rates (Supplementary Figure 3). Hence, we have kept the same learning rate as chosen in the original simulations. In the SNU experiments, small variations of the learning rate did not affect the final performance. Finally, we've varied the learning rate for the simulations with hardware constraints and verified that the improvements enabled by GRAPES are robust across different values of learning rate.

21, Referee: - *L453ff: Please elaborate how the difference of weight distributions is related to the number of epochs. I would not agree that the difference is necessarily smaller for fewer epochs.*

Response: We thank Reviewer #1 for the comment. We agree that the hypothesis that GRAPES is able to mitigate catastrophic forgetting because it achieves a lower training error in fewer epochs might not be always true. Indeed, GRAPES enhances the updates of some parameters, therefore the weight distribution can change in a smaller number of iterations compared to the standard SGD. We have removed this claim from the manuscript. On the other hand, we have elaborated on another intuition that can explain why GRAPES mitigates catastrophic forgetting. As shown in Supplementary Note 4, the weights learnt with GRAPES in some layers result in a distribution exhibiting a larger variance than the weights learnt with SGD. We speculate that such distribution might be more robust to performance degradation when the network is trained on a sequence of tasks. We elaborate on this intuition in lines 582ff.

22, Referee: - *Fig. 7 b) right hand side: $n=0.001$ instead of $n=0.01s$?*

Response: We thank Reviewer #1 for spotting the typo. We have corrected it in the legend of the revised Figure 7b.

23, Referee: - *L551: I do not get the meaning of "relative", here.*

Response: We thank Reviewer #1 for the comment. We have corrected the text in lines 674ff with the more appropriate expression "depending only on".

24, Referee: - *Suppl. Note 3: Not only the noise on and range of synaptic weights of hardware synapses is limited, but often also their weights, e.g. see [2]*

Response: We thank Reviewer #1 for bringing the work of Pfeil et al., 2012 to our attention. We have mentioned the constraint of weight resolution and included this Reference in the Supplementary note 5.

25, Referee: - *Table S3: Two results for GRAPES are missing.*

Response: We thank Reviewer #1 for the comment. The empty fields for slowness indicate that, despite the model can achieve an above chance level accuracy, the test curve exhibits a non-monotonical trend, that cannot be fit meaningfully with the plateau equation. We have included this clarification in the revised caption for

Supplementary Table 3 (now Supplementary Table 4): “Empty fields in the accuracy columns indicate no convergence, while empty fields in the slowness columns indicate that the shape of the test curve could not be meaningfully fitted with the plateau equation”.

26, Referee: - *Table S7: Not clear what +- denotes, standard deviation over trial averages?*

Response: We thank Reviewer #1 for pointing out this unclear point in the manuscript. The accuracy for each run is computed as the mean of the test accuracy over the last 10 training epochs. The reported result is the mean and standard deviation over the accuracy of 10 independent runs. We have modified the caption of the Table (now Supplementary Table 8) to include this clarification.

We hope that these revisions address the comments and questions of the Reviewer and provide a more complete picture of our work on the GRAPES optimizer.

[1] Salimans, T., & Kingma, D. P. (2016). *Weight normalization: A simple reparameterization to accelerate training of deep neural networks*. NIPS 2016.

[2] Pfeil, T., Potjans, T. C., Schrader, S., Potjans, W., Schemmel, J., Diesmann, M., & Meier, K. (2012). *Is a 4-bit synaptic weight resolution enough?—constraints on enabling spike-timing dependent plasticity in neuromorphic hardware*. *Frontiers in Neuroscience*, 6, 90.

Reply to Reviewer #2's Comments:

1, Referee: *Overview of Learning in Deep Neural Networks using a Biologically Inspired Optimizer:*

The authors present an artificial neural network (ANN) optimization strategy that is inspired by learning that occurs in mammalian brains. In a nutshell, the authors modulate the backpropagation signal back through the ANN (each layer individually) by computing the "importance" that a single feature/node has on the learned task. Finding better, brain-inspired optimizers for machine learning is critical for the community as we incrementally work toward developing artificial general intelligence; however, the authors have not compared against recent state-of-the-art nor have they shown that GRAPES scales to larger problems. This makes it difficult for reviewers to measure the impact of the proposed work; and since this Nature Communications is a high-impact publication, I recommend rejecting the paper in its current form. Specific comments are below.

Response: We are grateful to Reviewer #2 for critically assessing our work and for providing constructive comments. We've addressed the reviewer's suggestions and we've commented on the related revisions to our manuscript.

2, Referee: *1. The authors need to elaborate how Equation 1 (the sum of the absolute value of the weights in a given layer "l" and a node "n") corresponds to importance for learning a given task.*

Response: The concept of node importance that we introduce in Equation 1 takes inspiration from the mechanism of synaptic integration and relies on grouping the connections in a specific layer based on the postsynaptic node. The weight distribution of incoming synapses gives information on how much each node is responsible for the propagation of the signal from the input to the output layer. We use the absolute value because we are interested in the average strength of the connections – *i.e.* how much the weights "amplify" the signal – independently from their excitatory or inhibitory nature. We remark that the importance defined in Equation 1 depends only on the intrinsic state of the network and not on the input value. Therefore, in the initial phases of training, the importance is not related to the task. However, as the network undergoes training, the weight distribution becomes related to the task which the network is learning, and consequently the importance of the nodes becomes specific to the task. We have modified our manuscript to clarify these points in lines 221ff and 229ff.

3, Referee: *2. In Equation 2, why is the modulation magnitude bound between 1 and 2? The authors say that it is empirical, but the authors should elaborate on this design choice.*

Response: We have extended our study to include different ranges for m . We've reported the new results in the Supplementary Note 1. As shown, the best choice for the range of m is $[1,2]$, as in the original manuscript. However, note that the wider range $[1,3]$ leads to similar improvements with respect to the standard SGD.

4, Referee: *3. In Equation 3, what is W_o ? I may have missed it, but I only see it defined in Figure 2a.*

Response: We thank Reviewer #2 for noticing this point. W_o is the weight update computed with the standard optimizer (e.g., SGD). We have revised the manuscript to indicate this in lines 258ff of the main text as well as in the description of Figure 2a.

5, Referee: *4. Figure 3 is confusing. It is not clear to me that the graphic provides any meaningful tie back to the main points made in Lines 258-293 of the text.*

Response: We thank Reviewer #2 for the comment. Firstly, in Figure 3a we show that, during training, the mean of the modulation factor per layer decays for some layers and remains almost constant for other layers. In the text we suggest that a consequence of such dynamics is that GRAPES may present a similar effect as introducing a learning rate decay schedule. Indeed, the modulation factor acts like a node specific learning rate.

Importantly, GRAPES has the advantage of spontaneously adapting to the training conditions because the modulation factor evolves with the weight distribution. Therefore, GRAPES does not require to search and set any hyperparameter for the decay. Secondly, in Figure 3b we have showed that at the end of training the modulation factors have different distributions in terms of mean and variance for different layers. As a consequence, in the text we have highlighted that GRAPES allows to modify the decay schedule differently for each parameter, both within each layer and across layers. Finally, we have introduced the stability proof (in the Supplementary Note 2), which is necessary to ensure that the dynamic modulation scheme leads to convergence. We have modified the text in lines 315ff,333,340,346 to clarify which points in the text refer to which panels in the Figure. We hope that these changes clarify the aforementioned points adequately. Clearly, we would be happy to further modify the text should the reviewer have further concerns regarding clarity.

6, Referee: 5. *The experiments are “toy” examples that do not demonstrate the scalability of GRAPES. For example, on Line 407, the Lifelong Machine Learning (LML) community has replaced the MNIST data permutation experiment with larger scale problems such as ImageNet and Core50. I recommend checking out the REMIND paper located at <https://arxiv.org/pdf/1910.02509.pdf> for a scalable example, but there have already been improvements on this model.*^{SEP.6} *The authors must compare their results against the state-of-the-art algorithms, which I believe is necessary for high-impact publications such as Nature Communications. REMIND listed on the bullet above is an example of that.*

Response: We thank Reviewer #2 for bringing the REMIND algorithm to our attention. We have included references on the REMIND approach and on the general replay method in lines 496ff. REMIND modifies the standard training of a network to compress and store hidden representations of the model and to later augment the data during replay. In contrast, GRAPES is an optimizer and as such it does not introduce any additional steps to store nor replay samples during training, as it only modifies the magnitude of the learning rate differently for each parameter based on the weight distribution.

Moreover, GRAPES, unlike replay-based techniques, has not been designed to overcome catastrophic forgetting. We have empirically observed that GRAPES presents the benefit of mitigating performance degradation in simple tasks such as the MNIST permutation protocol. Therefore, we feel that a comparison of GRAPES against more sophisticated algorithms specifically designed for continual learning, such as REMIND, would not be fair, and we do not envision nor claim that GRAPES could compete with these state-of-the-art algorithms that introduce some form of replay. We remark, however, that GRAPES optimizer is independent of the algorithm and can be also incorporated in complex algorithms such as REMIND.

Regarding large-scale problems, please note that in the context of standard learning (not continual learning), we have extended our manuscript to demonstrate the improvements of GRAPES on datasets such as CIFAR-100, as reported in Table 1.

8, Referee: 7. *The LML Community is working to adopt a standardized benchmark located here: <https://avalanche.continualai.org/>. Especially for the toy examples, I strongly recommend using this framework to help other researchers provide apples-for-apples comparisons.*

Response: We thank Reviewer #2 for the constructive suggestion. We have extended our study to include results obtained through the Avalanche benchmark and demonstrated that, also with this benchmark, GRAPES is able to mitigate the performance degradation observed with the standard optimizer (SGD with momentum).

We have used the exact same protocol for the Permuted MNIST suggested in the library’s tutorial. The settings of the library are similar to the results we presented in the original manuscript, but some important differences exist. First, in Avalanche, there is no overlapping set of pixels between the tasks (*i.e.*, all pixels are permuted), while in our custom benchmark we permuted only 600 or 200 pixels out of 784. Secondly, in the Avalanche tutorial settings, momentum is used, which speeds up training and achieves above 90% accuracy per task in one training epoch only.

Importantly, our task setup involving a partial overlap between the tasks allows to explore further aspects of catastrophic forgetting compared to Avalanche, namely the utilization of partial information from previous tasks

and the per-task-future-accuracy. Therefore, we decided to include both the results with Avalanche and those obtained with our custom task setup. The new results obtained with the Avalanche library are shown in the revised Figure 6, panels a and b, whereas the original results obtained with our task setup are shown in panels c to f.

9, Referee: *8. The supplementary material provides mean + standard deviation over a set of five runs. The author needs to provide statistical significance testing for the different methods and ablation studies used in this paper. This will help reinforce the importance of the proposed work, push the author to increase the sample size, or make modifications to GRAPES until statistically significant improvements are demonstrated.*

Response: We thank Reviewer #2 for the constructive feedback. In the main manuscript, as shown in the revised Figures 4 and 5, we have extended our results to report a fine-grained ablation study of the learning rate. Furthermore, we increased the sample size to report the mean over a set of ten runs. We have also studied modifications of the magnitude of the modulation factor. As described in the Supplementary Note 1, we have varied the range of values of the modulation matrix and empirically shown that the range [1,2] leads to the best improvements.

We hope that these revisions address the comments and provide a more complete picture of our work on the GRAPES optimizer.

Reply to Reviewer #3's Comments:

1, Referee: *"Learning in Deep Neural Networks Using a Biologically Inspired Optimizer" reports a modification to the training procedure of artificial neural networks that integrates elements of a form of homeostatic plasticity observed in brain tissue. The modification is shown to lead to increased accuracy, greater learning speed and decreased vulnerability to catastrophic forgetting. Although a convergence proof is given in supplementary material, the main claims rely on computational simulations performed on benchmark machine learning tasks.*

There is currently quite a lot of interest in spotting subtle features of real neuronal network that would alleviate some of the outstanding issues with training deep artificial neural networks. Despite this strong interest, successes are rare. Potent and truly bio-inspired improvements have been difficult to show. The present article is squarely situated in the focus of this lens, but for the achievement to be valid we have to be careful that the improvements observed arise from a fair comparison and that the modifications are truly biologically-inspired. My concerns are that these two points are not addressed in a convincing fashion in the present form of the manuscript.

Response: We would like to thank Reviewer #3 for critically assessing our manuscript and providing constructive feedback. We have revised the manuscript according to the Reviewer's suggestions. In particular, we have revised our work to provide a more extensive comparison with SGD by including a finer grid search of the algorithms' dependence on the learning rate. Furthermore, we have enhanced the manuscript with a review of additional literature to relate the underlying concepts of the GRAPES method with biological mechanisms of neural circuits. Below is a point-by-point reply to the Reviewer's comments.

2, Referee: *Main concerns:*

Fair comparison with SGD. The results shown in Fig. 4 are impressive as the SGD and GRAPES show opposite dependence on depth. But I find this figure is an unusual description of SGD. Sure SGD can lead to the sort of biphasic depth dependence shown in Fig. 4, but a monotonically increasing dependence such as that shown for GRAPES is certainly possible. For instance, Li and Sompolinsky have derived some conditions for when a depth increases vs decreases performance. A change in hyperparameter can change whether depth increase or decreases performance. There are quite a few hyperparameters here. Of note, the ones that were chosen empirically in the choice of the min/max function for defining m_l . It is easy to imagine that in switching from SGD to GRAPES, a switch in task statistics resulted such that depth became entirely beneficial. Importantly, SGD with a more elaborate meta-parameter search could also have given this improvement. The improvement may thus have to do more on the relative choice of hyper parameters than the training protocol. The description in methods being very scant on the hyperparameter search (it appears only the learning rate was systematically varied), one would need a better description for the rationale of the hyperparameter search and/or a proof that SGD is being treated fairly in that respect. Particularly given that SGD-based curves at odds with those in Fig. 4 have been reported many times in the literature. The results pertaining to slowness may be more robust as I don't know of any reports at odds with what is shown here (though it is seldomly investigated).

Similarly, the improvements shown in Fig. 5 and 6 are small. It becomes a question whether this improvement is robust to different network setups (on that note, it would be useful to define the error bars, many changes don't even seem to be significant as per these error bars). The results in Fig 6 in particular appear to be right in the error bars of one another and do not seem to have been statistically compared.

Response: We thank Reviewer #3 for providing critical suggestions on how to improve our study on the scalability properties of SGD with and without GRAPES. Following the above comments, we have carried out an extensive exploration of the hyperparameter space to investigate the dependency of the performance on the network's complexity. For BP as well as for FA and DFA, we have systematically varied the learning rate in a range leading to successful training. We trained models of increasing complexity in terms of both layer size and depth. We observed that learning rates ($lr=0.1$) larger than the ones used in the original manuscript ($lr=0.001$) allow for higher final accuracy than previously reported. However, when larger learning rates are used in training, the dependence of the accuracy on the network's depth is less pronounced compared to the smaller learning rates shown in the original manuscript. Instead, we noticed that, with constant depth, increasing the layer size leads to a meaningful increase in performance, both for standard SGD and for GRAPES. As Reviewer #3

pointed out, we verified that, with a more elaborate meta-parameter choice, an increase in network complexity can lead to an increase in performance also with standard SGD. Moreover, we have shown that, for optimized values of the learning rate, the overall increase in performance with increasing layer size is more pronounced when GRAPES is applied (see Figure 4b). The surface plot in the revised Figure 4c depicts the final accuracy after training fully connected models with 4 hidden layers on MNIST with BP as a function of the learning rate and layer size. For both SGD (blue) and GRAPES (red) the accuracy increases as the learning rate and the layer size are increasing. Importantly, for all settings GRAPES reaches a higher accuracy than SGD. We remark that the models are not able to learn for larger learning rates than the ones reported. Furthermore, using the same strategy, we have extended the results for the slowness. The revised Figure 4d reports the dependence of the slowness on the learning rate and layer size. It shows that, as the learning rate is decreasing, the convergence rate of standard SGD significantly degrades, especially for small layer sizes, while such degradation is robustly mitigated by GRAPES.

Regarding the error bars, we have clarified in the figures' and tables' caption that the error bars represent the standard deviation (Figures 3a, 4b, 6, 7a, Table 1, Supplementary Fig. 3, Supplementary Notes 1, 3, 7, Supplementary Tables 2-8).

Finally, we have varied the range of the modulation factor by increasing the upper bound and by removing the lower bound. As reported in Supplementary Note 1, the ranges [1,2] and [1,3] lead to the best performance of GRAPES.

3, Referee: *Biological inspiration. The present algorithm is inspired by an observed property of real synapses that when potentiating one synapse, other synapses onto the same neuron are depressed. Although this is property is debated (see Beique et al. PNAS 2011), this type of constraint has long been an integral part of learning in biological neural network (see Oja's learning rule and PCA starting in 1982, up to more recent studies like Bird and Cuntz BioRxiv 2020). The approach taken here is different, however, as it does not normalize weights by the total input weight (as in Oja's rule), but by the *maximum* total input weight among the different units in a layer. As such, this introduces a form of non-local influence of synaptic weights. As if the neuron receiving the most weight is acting on all other neurons by dividing its total weight out of all synapse onto other neurons in the layer. To be specific, I am referring to Eq 2., which does look like the softmax function commonly used in conjunction with a reference to the biological winner-take-all computation implemented by feedback inhibition. But Eq 2 is on weights and neurons do not communicate weights so the feedback inhibition cannot apply. The authors mention that the same principle of homeostatic plasticity apply by considering Eq. 6, which is utterly confusing to me as I dont see how this is equivalent to Eq 2, nor how biology can do Eq 6. Together, I do not see how the core computational element studied here can be implemented by neurons.*

Response: We thank Reviewer #3 for the constructive feedback and suggestions. We agree with the Reviewer that, although our algorithm is reminiscent to existing normalization schemes and to the Winner-take-all computational primitive, it exhibits fundamental differences. Specifically, the normalization is based on the strongest average weight connection rather than on the highest activity. This is one of the novel elements that we introduce with our work. Indeed, as described in the manuscript, we distinguish the concept of network-driven responsibility, which is dependent on the intrinsic state of the network (*i.e.*, the weight distribution) from the concept of input-driven responsibility, which is dependent on the input instances. GRAPES relies on the network-driven responsibility, and therefore we designed the underlying computations, including the normalization operations, to be based solely on the weight distribution.

We agree that the normalization step of GRAPES described in Equation 2 implies that neurons communicate their weight strength. However, we introduce a normalization operation only to project the importance vector, whose values depend on the weight initialization, into a meaningful interval, in which values larger than 1 lead to enhancement of the weight updates, and smaller than 1 imply weakening of the updates. To clarify this point, we have added a diagram in the revised Figure 2b that illustrates the steps of normalization as well as the lower bounding in the simple case of three postsynaptic nodes. We would like to emphasize that we chose to perform the vector normalization by dividing by the maximum of the importance values, and to later multiply by 2 and lower bound by one, where the factor 2 has been empirically chosen to yield the best improvements (see Supplementary Note 1). We could have achieved the same numerical results by performing the vector normalization by dividing by the mean of the importance values (or the total sum of the importance values), and

later multiplying by $2 * \text{mean}(\text{importance}) / \text{max}(\text{importance})$ (resp. by $2 * \text{sum}(\text{importance}) / \text{max}(\text{importance})$) and finally lower bounding by 1. We also emphasize that the normalization does not act directly on the weight strength but on the modulation of the weight update.

Furthermore, previous research works have contemplated the possibility that the neurons communicate weights. For instance, in Fiete et al. (2010), the authors propose a network to model synaptic chain formation, which combines STDP with heterosynaptic long-term depression (hLTD). hLTD relies on the summed-weight limit rule: when the summed weight of synapses into (or out of) a neuron exceeds a limit, all the incoming (or outgoing) synapses to that neuron are weakened. Such mechanism implies that synapses communicate information on the weights to the postsynaptic node, and such information is used to modulate the synaptic strength of the incoming or outgoing synapses in a non-local fashion. The underlying mechanism of GRAPES is similar, with the difference that the weight modulation does not depend only on the total sum of synaptic strength but also on the maximum strength within the layer (for the normalization).

A second example that envisions communication of weight strength can be found in the theory of retroaxonal signals and neural marketplace proposed in [i, ii]. Experimental evidence suggests that neurons are capable of carrying retroaxonal signals through molecules known as neurotrophic factors, which can encode information both on synaptic strength and on its temporal derivative. Such information is used to promote or hinder the consolidation of synaptic changes. The theory of the neural marketplace builds on the mechanism of retroaxonal signalling and proposes a model for how networks of neurons in the brain self-organize into functional networks. Both the neural marketplace theory and the GRAPES algorithm rely on the propagation of information on weight strength and weight change, hence the two frameworks present several analogies. First, the retroaxonal signals control the plasticity of synapses by modulating the synaptic updates. Similarly, the importance vector is used in GRAPES to modulate the weight changes prescribed by BP. Secondly, the retroaxonal signals carrying information on weight strength and weight change travel slowly; similarly, the information in GRAPES is only applied after each batch. Third, both the information propagated through neurotrophin and the importance in GRAPES do not depend on gradients. Finally, the theory in [iii] introduces the concept of the worth of a cell, which measures the usefulness of its output, and is defined as the worsening in network performance if the cell were to die. A cell is inactivated if all its incoming connections are zeroes. Therefore, the worth of a cell is related to the strength of the incoming synapses to the cell. Thus, the worth of a cell can be related to the concept of importance in GRAPES.

Regarding Eq. 6, summing the connection strength on the outgoing synapses rather than on the incoming synapses is a modification necessary to adapt the algorithm to the DFA scheme, therefore the two schemes are not equivalent.

We have extended our manuscript to mention the above points in lines 692ff.

We also thank the Reviewer for bringing to our attention additional references, which we have included in the revised manuscript in lines 693ff.

[i] K. Harris, 2008

[ii] Lewis and Harris, 2014

4, Referee: *Other criticism to improve manuscript.*

Otherwise, I thought the manuscript structure difficult to follow.

Response: We thank Reviewer #3 for the feedback. We have restructured the introduction and the initial parts of the results. We hope this makes the structure easier to follow.

5, Referee: *The interwoven ideas of improving biological realism of ANNs and shining insight as to how to make biological simulations closer to ANNs was difficult to follow.*

Response: We thank Reviewer #3 for the comment. The main idea of our work is to incorporate mechanisms experimentally observed in biological circuits into the training of ANNs. Therefore, our work mainly aims to enhance the training of ANNs with algorithms inspired by biological mechanisms, rather than making biological simulations closer to ANNs.

6, Referee: *I was also confused by the two introductions.*

Response: We thank Reviewer #3 for the feedback. We have restructured the manuscript to include a longer introduction that incorporates the contents of the paragraph *Principles of biological computation* of the original manuscript. We hope that this restructuring makes the structure clearer.

7, Referee: *The title is not specific as to what element is the focus of biological inspiration.*

Response: We have changed the title to

Introducing Principles of Synaptic Integration in the Optimization of Deep Neural Networks.

8, Referee: *There were a number of inaccurate statements ('the stronger the input, the higher the firing rate and thus the amount of information propagated to the next layer', 'incorporates key principles of synaptic integration observed in dendrites' (should be plasticity not integration)), and as hinted at in the above points, the links with existing literature both biological and ML literature could be incorporated better.*

Response: We thank Reviewer #3 for the constructive suggestions. We have addressed them as follows:

'the stronger the input, the higher the firing rate and thus the amount of information propagated to the next layer': we have modified the text in lines 170ff to clarify that we are referring to rate-based models

'incorporates key principles of synaptic integration observed in dendrites': in the abstract we have replaced "integration" with "plasticity".

We have added additional references to deep learning literatures, e.g., in lines 30ff, 495ff, and to biological literature, e.g., in lines 170,694ff,721ff.

9, Referee: *I did not feel that Fig 1a was communicating the main point of the training protocol. Fig 1b was missing the effect of Eq 2.*

Response: We thank Reviewer #3 for the feedback on Fig 1. The aim of Fig 1a is to illustrate that the spatial distribution of presynaptic cells - and in particular their grouping in dendritic branches – influences the generation of dendritic APs, and consequently the impact that each presynaptic cell has on the postsynaptic axonal AP. We have expanded the text in lines 48ff to clarify the difference between dendritic and axonal spikes, providing links with Fig. 1a.

Furthermore, we have modified Fig 1b to include the effect of Eq. 2. In the revised figure, we have included dotted arrows emanating from the node with high importance and arriving to the other nodes in the layer. In this way, we have tried to indicate that normalization is performed based on the highest importance value within the layer. We have also added subfigure b in Figure 2 that shows a diagram illustrating the effect of normalization and lower bounding in the simple case of three postsynaptic nodes.

We hope that these revisions address the comments and provide a more complete picture of our work on the GRAPES optimizer.

REVIEWERS' COMMENTS

Reviewer #1 (Remarks to the Author):

Thank you for the detailed rebuttal. I still recommend the acceptance of this manuscript, but would like to ask you to consider the following points:

- The 3D plots are hard to read, especially Fig. 5 a and c. Consider alternative visualization methods, e.g. heat maps with identical value ranges.
- Your baseline performance for ResNet9 is 7% lower than the reference implementation on github that you refer to. This is a huge gap and certainly weakens your results on the CIFAR datasets.
- Related to point 15: I was addressing the last line of caption in Fig. 2: "... since δa_2 contains δa_1 ..."

Kind regards,

Thomas Pfeil

Reviewer #2 (Remarks to the Author):

The authors have satisfactorily addressed the concerns provided in my initial review. I am comfortable with accepting the article.

Reviewer #3 (Remarks to the Author):

Having now read the latest revision of the manuscript by Delaferrera et al., I can commend the efforts with which the authors have rather thoroughly solidify the comparison between their new learning algorithm and previous benchmarks. In my opinion, this addresses the main problems that I and other reviewers had about comparing algorithms without meta-parameter searches. The addition of lifelong learning protocols is also very nice. I think we can safely conclude that their algorithm does bear the small to moderate improvements in learning efficiency.

I had raised the issue that the algorithm is in fact not biologically inspired. This is a central statement of the article. In this regard, the revision is presenting a picture that is worse than the one in the original manuscript. In the new manuscript, we read first a statement that their algorithm is inspired by biology, then a presentation of an algorithm that does not fit with known biology, and then a discussion of how biology could in fact somehow do like their artificial algorithm. This is in my opinion an inconsistent setup. Is this a subtlety that will be missed by most readers? I don't know for sure, but I note that we are talking about 'weight transport', which is one of the most well known inconsistencies between artificial nets and biological nets. The manuscript still needs to correct its statement regarding biological plausibility.

Another minor point, as for Reviewer 1, I was also confused about the role of dendritic spikes in the whole story. These appear in Figure 1 but do not seem to be very important for the paper. And the new manuscript did not make this clearer for me. I thought that restricting the weight transport to within one neuron was perhaps the reason, but then much of the discussion about weight transport was talking about anterograde transport etc. So I am again confused.

The manuscript imply that their algorithm is not saturating as the number of trainable parameters grow. This is inaccurate and should be corrected.

Reply to Reviewer #1's Comments:

1, Referee: Reviewer #1 (Remarks to the Author):

Thank you for the detailed rebuttal. I still recommend the acceptance of this manuscript, but would like to ask you to consider the following points:

Response: We are grateful to Reviewer #1 for the positive feedback and for recommending the acceptance of our work. We also thank the Reviewer for critically assessing our manuscript once more. Below is a point-by-point reply to the Reviewer's comments. We have revised the manuscript accordingly.

2, Referee: *The 3D plots are hard to read, especially Fig. 5 a and c. Consider alternative visualization methods, e.g. heat maps with identical value ranges.*

Response: We agree with the Reviewer that the 3D surface plots are difficult to read. We have changed the visualization method to 3D bar plots. Figures 4 and 5 have been revised accordingly.

3, Referee: *Your baseline performance for ResNet9 is 7% lower than the reference implementation on github that you refer to. This is a huge gap and certainly weakens your results on the CIFAR datasets.*

Response: We thank Reviewer #1 for the comment. In our simulations, we have used a similar architecture as the one proposed in the referenced github repository, but we used custom simulation settings. In particular, as described in lines 495ff, we did not introduce weight decay nor weight normalization, as they introduce operations modifying the weights on which we compute the weight importance required to apply GRAPES. We leave for a follow up work an extensive exploration of the performance of GRAPES in combination with techniques such as weight decay and weight normalization.

4, Referee: *Related to point 15: I was addressing the last line of caption in Fig. 2: "... since δa_2 contains δa_1 ..."*

Response: We apologize for misunderstanding the Reviewer's comment and we thank the Reviewer for spotting the typo. We have corrected the caption of Fig. 2 in the revised version of the manuscript.

Reply to Reviewer #2's Comments:

1, Referee: *Reviewer #2 (Remarks to the Author):*

The authors have satisfactorily addressed the concerns provided in my initial review. I am comfortable with accepting the article.

Response: We thank Reviewer #2 for assessing our revised manuscript and for recommending the publication of our work.

Reply to Reviewer #3's Comments:

1, Referee: *Reviewer #3 (Remarks to the Author):*

Having now read the latest revision of the manuscript by Delaferrera et al., I can commend the efforts with which the authors have rather thoroughly solidify the comparison between their new learning algorithm and previous benchmarks. In my opinion, this addresses the main problems that I and other reviewers had about comparing algorithms without meta-parameter searches. The addition of lifelong learning protocols is also very nice. I think we can safely conclude that their algorithm does bear the small to moderate improvements in learning efficiency.

Response: We thank Reviewer #3 for the positive comments and for providing further constructive feedback on our work. We have addressed the Reviewer's suggestion below and revised our manuscript accordingly.

2, Referee: I had raised the issue that the algorithm is in fact not biologically inspired. This is a central statement of the article. In this regard, the revision is presenting a picture that is worse than the one in the original manuscript. In the new manuscript, we read first a statement that their algorithm is inspired by biology, then a presentation of an algorithm that does not fit with known biology, and then a discussion of how biology could in fact somehow do like their artificial algorithm. This is in my opinion an inconsistent setup. Is this a subtlety that will be missed by most readers? I don't know for sure, but I note that we are talking about 'weight transport', which is one of the most well known inconsistencies between artificial nets and biological nets. The manuscript still needs to correct its statement regarding biological plausibility.

Response: We thank Reviewer #3 for the comment. The underlying ideas of GRAPES are inspired by the concepts of node importance, error modulation and communication of weight strength, which are supported by experimental evidence from experiments investigating the role of dendritic integration, synaptic scaling and retroaxonal signalling. While the biological inspiration is grounded on these mechanisms, we agree with the Reviewer that the exact implementation of the importance-based error modulation proposed may not "fit with known biology". We added a paragraph in lines 759ff to clarify this and point out that only the high-level concept of GRAPES-like plasticity modulation are compatible with plasticity modulation principles observed in neural circuits. Furthermore, we have specified in lines 460ff that the propagating version of GRAPES incurs in the 'weight transport problem', also when applied to schemes such as DFA.

3, Referee: Another minor point, as for Reviewer 1, I was also confused about the role of dendritic spikes in the whole story. These appear in Figure 1 but do not seem to be very important for the paper. And the new manuscript did not make this clearer for me. I thought that restricting the weight transport to within one neuron was perhaps the reason, but then much of the discussion about weight transport was talking about anterograde transport etc. So I am again confused.

Response: We thank Reviewer #3 for the feedback on this point. Dendritic spikes represent a source of inspiration for the design of GRAPES. Indeed, they are an important example of how the spatial grouping of incoming input signals into dendritic branches modulates the impact of such signals on the postsynaptic cell. We have revised our manuscript to clarify the relationship between dendritic spikes and the definition of importance by adding a new paragraph in lines 197ff.

4, Referee: The manuscript imply that their algorithm is not saturating as the number of trainable parameters grow. This is inaccurate and should be corrected.

Response: We agree with this comment and thank Reviewer #3 for pointing this out. We have removed the sentence "better scalability of performance with network complexity," in lines 137ff and we have

modified the text in lines 407 and 668 to specify that GRAPES does not solve the saturation of the performance for increasing complexity, but it mitigates the issue when compared to SGD.